# Mapping ice formation to mineral-surface topography using a micro mixing chamber with video and atomic-force microscopy

Raymond W. Friddle[1] and Konrad Thürmer[1]

[1]Sandia National Laboratory, Livermore, 94550, USA

*Correspondence to*: Raymond W. Friddle (rwfridd@sandia.gov), Konrad Thürmer (kthurme@sandia.gov)

**Abstract.** We developed a method for examining ice formation on solid substrates exposed to cloud-like atmospheres. Our experimental approach couples video-rate optical microscopy of ice formation with high-resolution atomic force microscopy (AFM) of the initial mineral surface. We demonstrate how colocating stitched AFM images with video microscopy can be used to relate the likelihood of ice formation to nanoscale properties of a mineral substrate, e.g., the abundance of surface steps of a
certain height. We also discuss the potential of this setup for future iterative investigations of the properties of ice nucleation sites on materials.

## 1 Introduction

Ice formation[§] in the atmosphere initiates most precipitation and strongly affects Earth's radiation balance (Pruppacher and Klett 1997; Rogers and Yau 1989; Lohmann and Feichter 2005; DeMott et al. 2010; Lau and Wu 2003). Ice emerges via various
microscopic processes (Kanji et al. 2017; Pruppacher and Klett 1997; Rogers and Yau 1989; Vali et al. 2015); some require the presence of a foreign material (heterogenous nucleation) while others proceed unaided by any foreign substance (homogeneous nucleation). For ice nucleation to occur above ≈-36°C, a suitable ice nucleating particle (INP) must provide a surface onto which an ice nucleus can grow to a critical size without being impeded by an insurmountable activation barrier. Most ice nucleation events occur at atmospheric conditions where the critical-nucleus size ranges from ~1 nm to ~50 nm (Pruppacher and Klett 1997).
Uncovering the mechanisms involved in these events thus requires nanometer-resolution techniques.

While structure, morphology, particle size, and the presence of defects or functional groups have been found to determine the ice-nucleating ability of aerosols, the role of these properties and how they interact remains poorly understood (Coluzza et al. 2017; DeMott et al. 2011; Kanji et al. 2017; Koop and Mahowald 2013; Pruppacher and Klett 1997; Welti et al. 2014). Nevertheless,
recent ice-nucleation parameterizations, relating the concentration of INPs to the concentration of aerosol particles above a threshold size (DeMott et al. 2010), the aerosols' chemical composition (Vergara-Temprado et al. 2018), or water/substrate contact angles (Wang et al. 2014) and averaged field measurements, have succeeded in improving the accuracy of global climate models. However, as is true for extrapolations in general, such models are expected to be less accurate when applied to conditions outside the range of measured values used to fine tune these models, e.g., to predict a changing environment due to global warming or to
predict the behavior under extreme regional conditions, say, in plumes of dust or contamination. Developing a capability to predict ice formation at these uncharted atmospheric conditions with some confidence will require a quantitative understanding of the underlying mechanisms. Since macroscopic measurements of nucleation rates often cannot distinguish clearly between nucleation

---

[§] Throughout this manuscript we use the term "ice formation" for the entire process that produces ice, i.e., ice nucleation and growth and/or freezing.

mechanisms (Kanji et al. 2017; Marcolli 2014; Pruppacher and Klett 1997; Welti et al. 2014; Vali et al. 2015), innovative microscopy aimed at uncovering the local structural and chemical properties of ice nucleation sites is needed.

The technical requirements to adequately resolve ice nucleation in time and space are indeed demanding. The imaging system must operate in humid environments, at sub-0 °C temperatures, be non-destructive, and offer spatial resolution on the order of nanometers. Finally, to locate the ice nucleation site to within nanometers, the frame rate must be fast enough to capture the earliest emergence of the crystalline phase. As a point of reference, our data at $\approx$ -30 °C and relative humidity $RH_w \approx 100$ % suggest that capturing just one frame of a new ice crystal smaller than 10 nm would require imaging at rates greater than 1 Mfps (< 1 μs per image). This minimum frame rate decreases with lower temperature and humidity. Various groups recently demonstrated the resolving power of environmental scanning electron microscopy (ESEM) for studying ice nucleation (Kiselev et al. 2016; Wang et al. 2016; Zimmermann et al. 2008). Unfortunately, ESEM is unable to operate under realistic atmospheric pressures and is prone to introducing electron-beam damage and local heating of condensed water (Rykaczewski, Henry, and Fedorov 2009).

AFM is unique among high-resolution microscopies in that it non-destructively generates the 3-dimensional topography of a surface. Furthermore, the force-based imaging principle of AFM permits it to operate in a broad range of environmental conditions, including exposure to gases, liquids, and varied temperatures. This environmental versatility would appear to make AFM well-suited for *in situ* imaging of cloud-like icing processes on the surface of particulates or other samples. However, a key limitation of most commercial AFMs is their slow imaging speed. Current developments in high-speed AFM (HS-AFM) are encouraging, and have the potential to significantly advance heterogenous nucleation research (Ando 2014; Russell-Pavier et al. 2018), if emerging ice crystals could be observed directly with ~10 nm resolution, thus improving significantly the accuracy of locating nucleation sites.

By combining the speed of optical microscopy with the spatial resolution of AFM, the limitations of the individual instruments can be mitigated by colocation. Here we present an approach to connect optical images of ice forming locations to AFM data that resolve mineral substrate surface structures at the nanometer scale. While studying aerosol particles collected from the atmosphere would provide a more direct connection to atmospheric conditions. The typically complex structure and chemistry of these particles often precludes identifying the individual nanoscale processes that are important. For this study, instead, we choose extended flat substrates of known composition on which the role of individual topographic features can be examined. We demonstrate our approach on a substrate of K-feldspar (orthoclase), where we observe and quantify how surface steps facilitate ice formation– a phenomenon pertinent to ice nucleation and growth in mixed-phase clouds.

## 2 Experimental apparatus and methods

### 2.1 Setup of a small mixing chamber AFM with video microscopy

#### 2.1.1 Overview of AFM, video, and gas flow components

The experimental setup (Figure 1) is built around a Multimode 8 AFM operated by a Nanoscope V controller (Bruker, Santa Barbara, CA). The AFM is seated on a Nikon top-down microscopy stage equipped with a 10X long-working-distance lens (WD = 49.5mm, NA = 0.2, resolution approximately 1.6 μm). Video microscopy is recorded with an Infinity3-3URC 2.8 MP, 53 fps, color CCD camera. The AFM scanner, head, and objective lens are maintained in a dry nitrogen environment by a cylindrical

acrylic atmospheric hood (MMAH2, Bruker). Thermally conductive epoxy (KONA 870FT LVDP, Henkel) is used to adhere the sample to a glass slide which is glued to a copper standoff stage. The standoff stage creates open space between the sample and the AFM piezo for underside cooling and placement of a thermistor.

Cold nitrogen gas is used for cooling both the substrate and the gas above the substrate. Ultrapure nitrogen gas, initially at room temperature, flows through a heat exchanging copper coil immersed in a dewar of liquid nitrogen. The final temperature of the cooled nitrogen is controlled by mixing with room temperature nitrogen. The cooled nitrogen is then divided into two paths: one cools the underside of the copper standoff stage, the other enters an inlet of the AFM sample cell. Water vapor is generated by flowing ultrapure dry nitrogen through a water bubbler. The humidity of the resulting room-temperature vapor is measured using
a humidity sensor (ThermaData Series II – HTF, ThermoWorks) with an accuracy of $\pm3\%$ $RH_w$ at 25 °C. This vapor is then piped directly into the second inlet of the AFM sample cell. The temperature of the sample cell is monitored by thermistors placed at the underside of the copper standoff and the outlet of the sample cell. The thermistor readings are sampled at 1 Hz using a TC-720 thermoelectric temperature controller (TE Technology, Inc.). With this setup we are able to adjust the room-temperature $RH_w$ between 0 – 90 %, and the temperature of the sample stage to as low as -70 °C.

**2.1.2 Micro Mixing Chamber**

To create a cloud-like atmosphere in a small-volume sample chamber requires humidity near saturation ($RH_w \approx 100\%$) at sub-zero temperatures (< 0 °C). This is difficult to achieve in practice since delivering water vapor through a small tube at freezing temperatures will inevitably clog the tube with ice. Therefore, the vapor must remain above freezing temperatures during transport to the sample cell, then immediately cooled to the desired temperature. To achieve this combination of cold and humid gases we
use a glass AFM fluid cell with three ports (Bruker, model ECFC): two ports located next to each other deliver the gases which combine upon entry into the sample chamber, while the third port serves as an outlet. The port arrangement is shown in Figure 1a. This configuration separates the sub-zero dry nitrogen from the water vapor until reaching a small mixing column at the entry into the sample chamber. The volume of the sample chamber is approximately 30 µL, enabling rapid exchange of gases.

As mentioned in the previous section, the sample is cooled by the cold $N_2$ gas entering the chamber and by cold $N_2$ gas impinging on the underside of the sample. In our experiments, 0.66 L/min of the cold $N_2$ flows into the chamber, while 13.44 L/min of cold $N_2$ is directed at the underside of the sample puck. Measurements of the temperature of the gas exiting the chamber find it to be about 5 to 6 C warmer than the reading directly under the sample puck. Some of the exit gas warming arises from exchanging heat with the small diameter outlet tube that the gas passes through before contacting the sensor (Figure 1a).

A number of advantages come with the mixing-chamber approach to observing ice formation. Since the entry gas is already at the desired temperature, a range of flow rates can be explored (here 0.9 – 1.32 L/min), without concern for cooling by interaction with a cold surface. In a cold stage approach, measures must be taken to ensure that the coldest part of the cell volume is the sample under study (Wang et al. 2016), otherwise water will condense on unwanted components. In our setup, a cold atmosphere flows
into the small ~ 30 µL volume cooling the sample from above and below, thus minimizing temperature differences laterally and from the sample-gas interface upwards. The last point has the potential benefit of keeping the AFM tip near the same temperature as the sample surface, allowing, in principle, to image ice or the sample surface without raising their temperature (not explored here).

## 2.2 Experimental procedure

First, we pre-record detailed maps of the mineral-surface morphology with AFM at room temperature using a Tap150Al-G probe (Budget Sensors) without introducing humidity. To be able to capture a relatively large surface area of typically ~750 × 570 μm² while maintaining sub-nanometer height resolution, we developed an AFM stitching procedure described in Section 2.3.

Subsequently, we performed ice growth experiments by exposing the same mineral surface region to a cold and humid environment at atmospheric pressure. Here, we optically monitor the icing process at a set temperature while switching on and off the flow of water vapor. Following the diagram in Figure 1, a single source of compressed nitrogen supports three primary gas pathways: the humidifying bubbler, the warm mixing line, and the LN₂ cooling dewar. The sample temperature is set by adjusting the mixing ratio of room-temperature nitrogen to LN₂-cooled nitrogen until the sample thermistor reads the desired temperature. The

temperature is not actively controlled, therefore a small change in the inlet gas temperature will occur when the vapor is mixed into the stream. During the course of an icing experiment the temperature measured at the chamber outlet warms by about 1 °C , while the sample stage cools by approximately 0.5 °C due to the diverting of a portion of cold gas back to the stage when the vapor flow is turned on. Most of the cold dry nitrogen stream is delivered to the base of the sample holder, while a small fraction is directed to one of the sample cell inlets. During temperature settling, a 3-way diverting valve passes the humidified nitrogen gas

exiting the bubbler through a humidity sensor to measure its relative humidity. This maintains a dry, cold sample chamber while the vapor stream reaches a steady-state humidity. To inject water vapor into the cell, the 3-way valve above the bubbler is switched to divert the vapor stream into one of the mixing inlets of the cell. The vapor mixes with the steady stream of cold gas to supply cold vapor to the sample chamber. To halt the experiment, the vapor is again diverted to the humidity sensor. The ice is then removed from the surface before the next experiment through sublimation under dry N₂ and increasing the temperature above 0 °C.


Absolute humidity values ($AH_{in}$) provided hereafter represent estimates of the water content of the gas stream injected into the environmental chamber. Note that after condensation and especially ice formation has started, the atmosphere in the microliter environmental chamber enters a non-equilibrium stage, in which the local humidity, especially near growing ice features, is much lower than the given $AH_{in}$ values. We estimate $AH_{in}$ by calibrating against the saturation ($RH_w ≈ 100$ %) condition. To establish

the $RH_w ≈ 100$ % condition, we adjust the humidity inlet flow until we see droplets of ~1-5 μm diameter neither grow nor shrink, until after a few seconds the first ice crystals appear, causing nearby droplet to shrink and disappear. We then calculate the absolute humidity at T = -29.5 °C and $RH_w = 100$ % to find $AH_{100} = 0.48$ g/m³ at a flow rate of $Q_{100} = 0.28$ L/min, where $AH_{100}$ and $Q_{100}$ represent the absolute humidity and flow rate *at saturation* for that temperature. For constant dry cold flow rate, we assume the fraction of humid gas incorporated into the total inlet gas to be linear over the flow rates employed here (0.28 – 0.66

L/min). Therefore, for a humid line flow rate, $Q$, the humidity injected into the cell is approximated by,

$$AH_{in} = AH_{100}\frac{Q}{Q_{100}}.$$

We estimate that a 0.05 L/min error exists in our measurement of flow rate leading to an error for $AH_{in}$ of ±0.08 g/m³.


## 2.3 AFM image stitching

Several AFM images covering a relatively large area of the feldspar surface were acquired on a Dimension 3100 (Bruker), with a Nanoscope V controller, in tapping mode. This AFM has a motorized XY stage that allows programming a grid of images to be

acquired at locations that cover the desired surface region. A mosaic is produced by stitching together a 6x9 array of 54 individual AFM images, each with a $100 \times 100~\mu m^2$ scan size and typical overlap of 10 μm with neighboring images. The large scan size and acquisition time result in appreciable background warping of the individual images. To optimize stitching of adjacent images with minimal seam lines requires flattening each image. In most cases we subtracted a 2D polynomial of first order in $x$ and second order in $y$, $\sum_{j=0}^{1}\sum_{k=0}^{2} a_{j,k} x^j y^k$, which is fitted to masked areas of constant height. The resulting image is then levelled by an iterative routine which optimizes levelling of surface facets (software: Gwyddion). Image borders are seamed by alpha blending such that the image height, $h_i$, across the overlap of images $i = 1,2$ is blended by $h_{tot} = \alpha\, h_1 + (1 - \alpha)\, h_2$, where $\alpha$ varies linearly from 0 to 1 across the width of the overlap. Note that quantitative analysis of step heights is performed on the original individual images to avoid errors caused by seams. Image alignment and blending is performed using a custom routine in Igor Pro (Wavemetrics).

## 3 Experimental results

We used the system described here to examine ice formation on a sample mechanically cut from single crystal K-feldspar along the (001) easy-cleavage plane (Orthoclase, $KAlSi_3O_8$, Yavapai County, Arizona, USA, vendor: VWR/Eric Miller). In this study we maintain a fixed temperature of approximately –29.5 ± 0.5 °C while recording video of ice formation at different humidities. At humidity below saturation ($AH_{in} = 0.40 \pm 0.08$ g/m$^3$), we find growth of isolated ice crystals on the feldspar surface. This observation would be consistent with the classic mechanism of deposition nucleation, where ice nucleates directly from vapor without prior formation of liquid (Vali et al. 2015). However, as Marcolli (Marcolli 2014) has pointed out, many observations that appear to be deposition could actually be pore condensation and freezing of water, stabilized in cavities at $RH_w<100\%$ due to the inverse Kelvin effect (Christenson 2013; David et al. 2019; Fukuta 1966; Marcolli 2014; Pach and Verdaguer 2019). Since such capillary condensation could have occurred on confined surface structures just a few nanometers wide, optical microscopy would not have been able to detect it. Near saturation with respect to water ($RH_w \approx 100\%$, $AH_{in} = 0.48 \pm 0.08$ g/m$^3$), we observe condensation of water droplets on the K-feldspar surface (Fig. 2f-h). Ice formation at various sites occurs either concomitantly with condensation or after a short induction period. Alongside the optical images of ice formation in Fig. 2 are AFM images of the same locations. We find that many distinct surface sites repeatedly nucleate ice across multiple experiments, and over varied humidities, which agrees with ESEM findings of active sites on orthoclase for heterogeneous ice nucleation (Kiselev et al. 2016). However, we also observed that after covering the surface with liquid water, then subsequently drying the surface before repeating an ice experiment, some sites lost their ice nucleating ability while previously inactive locations became sites for nucleation.

Above saturation, we observe a very different pathway to ice formation. As shown in Fig. 3, the initially dry surface is first darkened by the condensation of water droplets on the sample surface. Shortly thereafter, rough filaments of ice branch out across the surface. A denuded zone is established as a halo absent of water droplets around these ice filaments. After the elongation of the filaments halts, the width of the ice filaments continues to grow as water from the surrounding vapor attaches to the crystals. Qualitatively comparing the optical images to the colocated AFM image in Figure 3, it appears that the ice filaments follow the contour of surface step edges. In Figure 4 we show that this is indeed the case. There, a frame of optical data taken when the ice filament extensions have ceased (Fig. 4b), is overlaid on a mosaic AFM image of the same area (Fig. 4a) to produce a colocated composite (Fig. 4c). Clearly, the ice decorates many of the prominent step edges on the surface. The emergence of these ice-filament patterns has been described in more detail in (Friddle and Thürmer 2019a). We also find a few isolated ice crystals (two are labelled in Fig. 4b), which despite having been surrounded by nearby droplets supplying water, did not merge with the main continuous system of

ice filaments. Comparing the surface structures underlying these isolated crystals to that for filaments in Figs. 4d-g, we see that for the surfaces where ice forms extended filaments the surface presents tall step edges that run uninterrupted along the extension of the ice filament's path (Figs. 4f,g). The substrate surfaces underlying the isolated ice crystals, on the contrary, display island-like protrusions spanning relatively short distances (Figs. 4d,e). Our data neither reveal nor rule out any preferred crystal orientation of the observed ice structures.

## 4 Data processing

### 4.1 Analysis of surface steps

Figure 5 illustrates the process we developed to extract step heights from the AFM data and relate those to the optically-observed ice-forming locations. An AFM image (Fig. 5a) is first converted to an image of step-heights (Fig. 5b), where traces outline where the step edges lie, and the value of each pixel along the trace corresponds to the step's height at that point. This step-edge image is generated by the following custom routine (Igor Pro, Wavemetrics) operating on the row and column 1D arrays of the 2D image matrix: After background subtraction, the derivative of the 1D array is taken. An edge is found when the rate-of-change of the differentiated array crosses a threshold of $d^2z/dx^2 = 6$ nm/mm$^2$, which detects step heights greater than 2 nm. Once a crossing is found, the local peak and two floor points of the derivative array are used to determine the step height from the original array. The step edge location and height are assigned to a pixel on the new image. This process is repeated, line-by-line, along the rows and columns of the AFM image.

The step edge image is interactive (Fig. 5b) to facilitate manually collecting step-height statistics over many step segments. Contiguous pixels of similar step height are grouped into clickable trace segments which change color and thickness to indicate selection (see Fig. 5d). Selection is reversible, and trace segments can be cut into smaller segments where needed to match the iced segment lengths observed optically.

### 4.2 Relating ice formation to step heights

Once registration between the AFM step height image and optical image is established, the routine discussed in section 4.1 is applied to collect statistics on the heights of step edges along which ice forms. As shown in Figure 5, the selected step edges in panel 5d are chosen to coincide with the filaments of ice in panel 5c. The selected trace segments contain step height values for each pixel along the segment. This is repeated for all the images across the desired analysis area. The selected step heights are binned into a histogram and compared against a histogram of all step heights presented in the image. Binning is counted as pixels or physical length.

Figure 6 shows an example of processed step height data collected over 15 images, each covering $100 \times 100$ μm$^2$. Here we show histograms for ice formation on K-feldspar at four different humidities, all at a temperature of $-29.5 \pm 0.5$ °C. Each histogram is derived from one video frame for each humidity which is chosen based on when ice propagation across the surface has halted. Figure 6 shows the distribution of step heights for a) all steps observed and b) steps on which ice propagates. The ratio of these two histograms – total length of iced steps within a step-height bin over total length of *all* steps (within the same step-height bin) – provides the probability of finding ice on a step of a given height (Fig. 6c).

# 5 Discussion and outlook

The data in Figure 6c reveals that, above saturation, ice is more likely to form along taller steps than shorter steps. Furthermore, the sigmoidal probability distribution in Fig. 6c shifts with changing humidity: as the humidity is increased, the curve moves towards smaller step heights. As detailed in (Friddle and Thürmer 2019a), the formation of ice filaments along step edges can be explained by capillary water condensation, with water filling the bottom corner where the step edge meets the underlying terrace. The orthoclase water contact angle is ~ 45° (Karagüzel et al. 2005), and thus, at saturation, perpendicular steps of all heights will be lined with water wedges when the surface's water contact angle ≤ 45° (Brinkmann and Blossey 2004; Moosavi, Rauscher, and Dietrich 2006; Seemann et al. 2011). Therefore, steps of any height are pre-filled with liquid water which can freeze in place once a nucleation event occurs anywhere along the step. This process can be viewed as an extension of the pore condensation and freezing mechanism (Christenson 2013; David et al. 2019; Fukuta 1966; Marcolli 2014; Pach and Verdaguer 2019) to higher humidity. A step height-dependence of ice growth can be attributed to two causes. First, if the probability of a given body of supercooled water to freeze at any given moment is roughly proportional to the surface area of the feldspar substrate immersed in the supercooled water, consistent with the models for immersion freezing considered in (Knopf et al. 2020), then the probability of a given step edge segment to initiate ice nucleation is roughly proportional to its height. The second cause arises from the subsequent dehydration of water wedges as denuded zones form around existing ice crystals. The taller steps contain more water, and thus can retain water wedges longer than their shorter neighbours. Hence the transformation of water wedges to ice, due to heterogeneous nucleation or contact with a nearby ice filament, has a greater window in time to occur with taller steps. This capillary-based mechanism is consistent with the ice formation shown in Figure 4. Here, ice filaments grow when step edges maintained tall heights for extended distances, whereas isolated ice crystals were observed at step edges having short lengths, such as protrusions or depressions surrounded by flat areas.

The surface region we analyzed quantitatively is smaller than 500 μm x 300 μm, hence the temperature differences within this area are expected to be small. As can be seen directly in the images of fig. 4, and the analysis in Fig. 6c, any possible temperature effects were not able to obscure the very strong correlation between ice-formation patterns and surface-step patterns. Hence, the rapid emergence and propagation of ice filaments is clearly not driven by temperature gradients, but rather governed by the surface-step patterns. The subsequent, much slower, expansion of the clustered ice crystals decorating the steps, which was not examined quantitatively, could in principle, be more affected by temperature gradients, although visual inspection of the optical images does not reveal strong evidence for this.

Typically, most aerosol particles are completely immersed in a cloud droplet already at very modest supersaturations. As discussed in detail in (Friddle and Thürmer 2019a), the step-facilitated mechanism described above is expected to be relevant when a cavity-free feldspar particle, initially devoid of ice, is suspended in air colder than -20°C that becomes slowly saturated. According to Fletcher's estimate (Fletcher 1962; Pruppacher and Klett 1997) that a humidity of $RH_w > 130\%$ is required for a measurable nucleation rate of water droplets with a contact angle of ≈ 45° on a planar insoluble substrate. Hence condensation of supercooled water will be confined to step edges, where the water will freeze rapidly, thus initiating ice formation.

In the discussed example of ice formation, the step-height analysis is used to corroborate the involvement of the liquid phase of water during the observed rapid formation and propagation of ice on feldspar, while the link between surface-step height and the ability of an isolated aerosol particle to initiate ice nucleation is neither direct nor obvious. Nevertheless, such step-height analysis

might benefit future studies in fields of material science, like corrosion and aircraft icing (Gent, Dart, and Cansdale 2000; Kreder et al. 2016), where the behaviour of the examined materials is affected by the abundance of surface steps.

In the current implementation of the AFM/optical technique presented in this paper, the two microscopies are performed sequentially on the same surface area, and the data are subsequently merged for quantitative analysis of surface structure as it pertains to ice formation. Due to the limited resolution of optical microscopy, we cannot directly determine the ice-nucleation site with nanometer precision. But by preparing cleavage surfaces that contain rather flat surface regions, we create configurations in which most steps are separated far enough to be optically resolved, allowing us to relate ice-propagation patterns to surface step patterns, and to employ AFM to quantify the role of the steps' height. Future experiments will explore simultaneous operation of AFM and optical microscopy, which may improve spatial localization of ice nucleation sites and resolving their morphologies. The setup described here is also applicable to studying deposition mode nucleation at sub -36 °C and sub-saturation (relevant to cirrus cloud formation), and optical observations of immersion mode nucleation on substrates. As previously demonstrated (Yang et al. 2015; Gurganus, Kostinski, and Shaw 2011; Yang et al. 2018; Holden et al. 2019), higher frame rates than used here are imperative when the sample is immersed in water because the ice can spread to millimetre length-scales on the order of milliseconds after the nucleation event. Nevertheless, higher video frame rates do not improve the resolving power of the microscope which is fundamentally restricted by the diffraction limit $\sim \lambda/2NA$. In our system the small numerical aperture of our objective (NA = 0.2) follows from the large working distance lens required to fit within the clearance of the AFM. Dedicated microscopy systems can improve resolution by using higher NA and by implementing blue filters to limit the wavelength $\lambda$. Ultimately, advances in highspeed AFM (HS-AFM) may be the key to direct observations of ice nucleation events. A typical commercial-AFM scan takes 10 seconds to 10 minutes depending on flatness of the substrate, the field of view (FOV), and the desired resolution. Meanwhile at $RH_w \approx 100$ % and $\approx -30$ °C we estimate that a new ice crystal reaches an effective diameter of 1 μm in just 6 ms[**]. Progress in HS-AFM development has proven the tool to be capable of performing fast imaging of dynamic processes at nanometer resolution under various environments (Yamashita et al. 2009; Payton, Picco, and Scott 2016; Pyne et al. 2009; Picco et al. 2008; Kodera et al. 2010; Uchihashi et al. 2011; Casuso et al. 2010).

Lastly, the rapid propagation of ice we see on feldspar, should also occur in other circumstances where extended surfaces, covered with continuous networks of steps or grooves, are exposed to a supersaturated air, providing a microscopy-based argument for avoiding surfaces with large steps or grooves in efforts to suppress aircraft icing (Gent, Dart, and Cansdale 2000; Kreder et al. 2016).

*Video supplement*. Videos corresponding to the optical frames presented in figures 2a, 2b, 3a, and 4b can be found online at https://doi.org/10.7910/DVN/DZUZ6P.

*Data availability*. Raw AFM and optical video data supporting the findings of this study are available from RWF (rwfridd@sandia.gov) or KT (kthurme@sandia.gov) on request.

*Author contributions*. RWF and KT conceived of and performed the experiments, analysed the data, and wrote the paper.

---

[**] Based on video observations of projected areal growth rate of an ice crystal of 130 $\mu m^2$/s treated effectively as a circle.

*Competing interests*. The authors declare that they have no conflict of interest.

*Acknowledgments*. We thank Norman C. Bartelt for insightful discussions. This work was supported by the Sandia Laboratory
Directed Research and Development Program. Sandia National Laboratories is a multi-mission laboratory managed and operated
by National Technology and Engineering Solutions of Sandia LLC, a wholly owned subsidiary of Honeywell International Inc. for
the U.S. Department of Energy's National Nuclear Security Administration under contract DE-NA0003525.

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

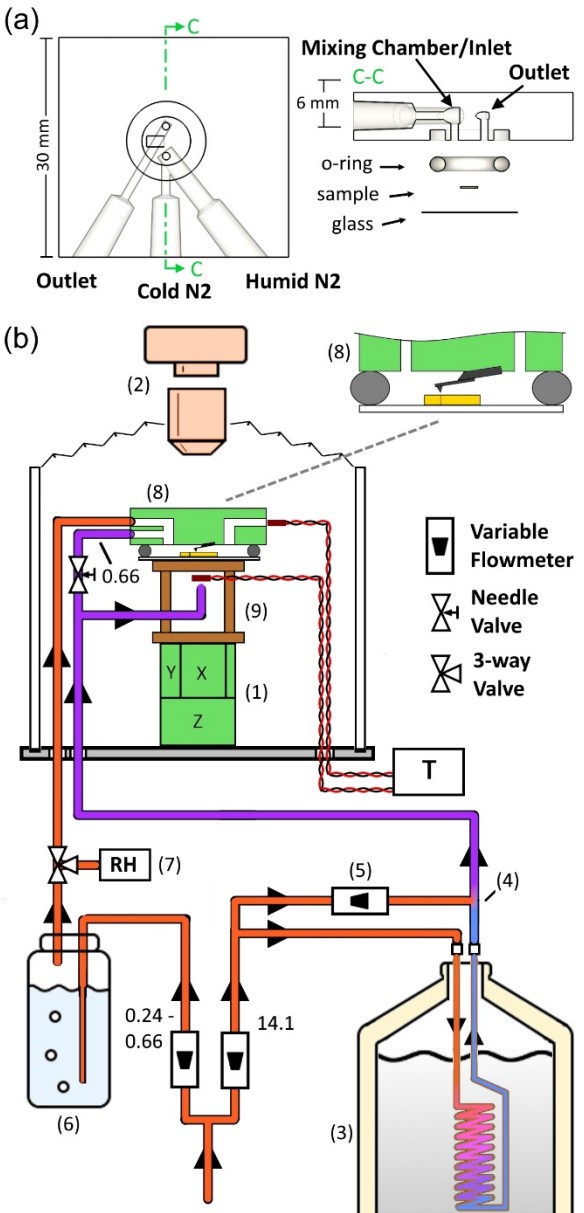

**Figure 1:** Experimental Setup. **(a)** The 3-port fluid cell. The port design separates the humid and cold gases before injection into the cell, which are then mixed to form a cold humid atmosphere just before entry to the sample volume. **(b)** The overall setup is built around an existing AFM (8) with top-down optics (2). Dry nitrogen is divided into cold and humid streams. The cold stream is nitrogen gas cooled in a liquid nitrogen-filled dewar (3), mixed at (4) with a dosage of room temperature nitrogen gas varied by (5). The total flow rate of the cold stream is unchanged, only the proportions of cold and warm flows are adjusted. The humid stream is generated by flowing nitrogen through a bubbler filled with water (6). The humid gas leaving the bubbler is directed by a 3-way valve towards either a humidity sensor (7), or to the sample cell. The cold and humid streams of nitrogen gas enter an acrylic bell jar which houses the AFM scanner with a cellophane bellows bridging the optical objective (2) to the top rim of the jar. The humid stream enters one port of the sample cell while the cold stream is divided to cool the underside of the sample, by way of a copper stand-off stage (9), and flow a smaller proportion into the other port of the sample cell. The temperature of the underside of the sample stage and the gas exiting the cell are measured with thermistors. The decimal values next to flow line segments are flow rates for those segments in L/min.

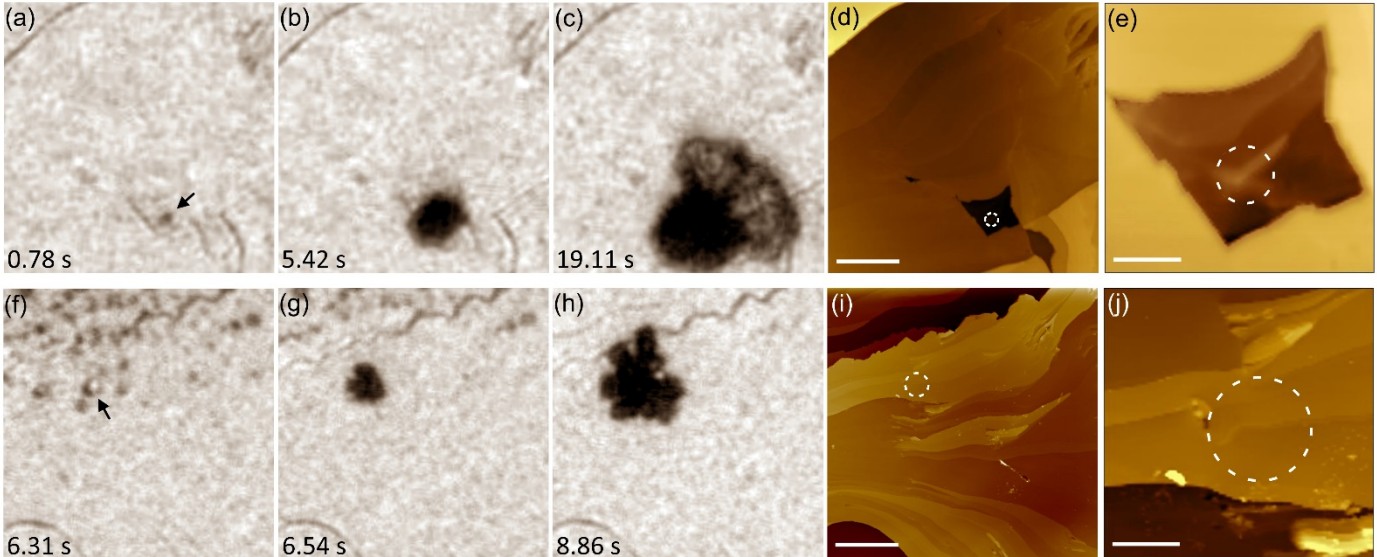

**Figure 2:** Colocating isolated nucleation events at -29.5 ± 0.5 C to AFM topography. **(a)-(c)** Three frames of an ice crystal nucleating and growing under $AH_{in} = 0.40 \pm 0.08$ g/m³. **(d)** AFM image of the same location in a-c) with dashed circle around the area of nucleation. Scale bar 20 μm. **(e)** Expanded view of a portion of panel (d) showing a small protrusion within a larger pit. Scale bar 5 μm. **(f)-(h)** Frames taken under $AH_{in} = 0.48 \pm 0.08$ g/m³ which show a collection of droplets on the surface in panel (f). The arrow points to the droplet which initiates an ice crystal. **(i)** AFM of the same area as in (f)-(h) with a circle around the original droplet in (f). Scale bare 20 μm. **(j)** Expanded view of panel (i). Scale bar 5 μm. Time stamps in lower left corner of optical frames are relative to the start time of their respective movies, however the time when water vapor fills the cell is not measured with significant precision. Accompanying videos can be found at (Friddle and Thürmer 2019b).

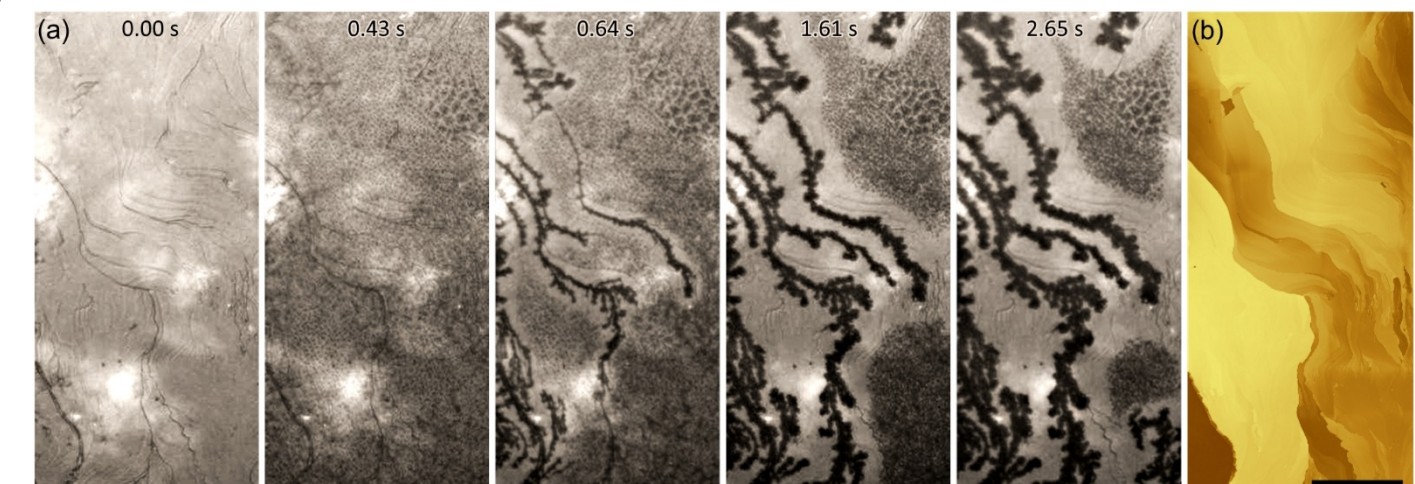

**Figure 3:** Ice growth on the feldspar surface at -29.5 ± 0.5 C and $AH_{in} = 0.80 \pm 0.08$ g/m³. **(a)** Video frames at the noted times show the progression from a dry surface (0.00 s), to surface water condensation (0.43 s), and finally to propagation of ice across the surface. From 0.64 – 2.65 s the denuded zone (dehydrated halo) around the ice filaments expands with ice growth. **(b)** AFM mosaic of the same area as in a) where lighter color indicates higher surface topography. Note the prominent step edges follow the same path as many of the ice filaments. Scale bar 100 μm. Accompanying video can be found at (Friddle and Thürmer 2019b).

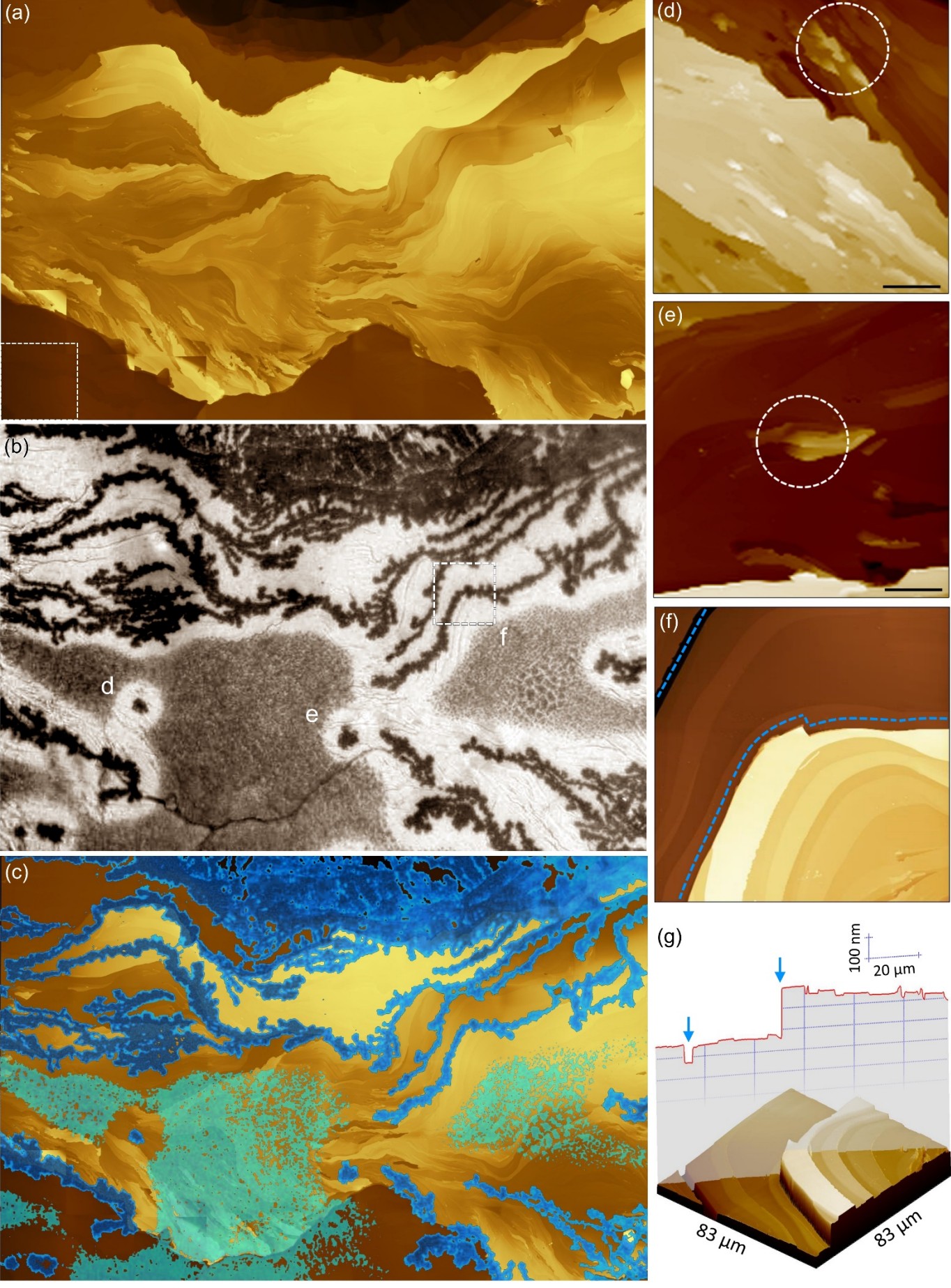

**Figure 4:** Wide field of view colocation of AFM data and optical video of ice growth at $-29.5 \pm 0.5$ C and $AH_{in} = 0.80 \pm 0.08$ g/m$^3$. **(a)** A mosaic composed of 54 individual AFM images stitched together to form a topographical map spanning $750 \times 570$ µm$^2$. The dashed box in the lower left represents the $100 \times 100$ µm$^2$ size of each individual AFM image. Brighter color represents higher topography. **(b)** An optical video frame captured over the same area as in a) showing ice growth. Three locations of interest are marked on the optical image (d, e, and f) and AFM images corresponding to their locations are shown in the panels at the right with the same labels. **(c)** The optical image in panel (b) is thresholded and false-colored, then overlaid on the AFM data shown in panel (a). The resulting overlay demonstrates the preference of ice formation along prominent step edges. **(d),(e)** Two example regions where ice nucleated on the surface and grew as isolated crystals without propagating across steps. Dashed circles enclose the locations where ice nucleation occurred. Scale bars are 5 µm. **(f)** Planar and **(g)** 3D view of the region marked f in panel (b). Dashed lines in (f) represent the path followed by ice, and the arrows in the height profile in (g) mark locations of ice formation. Video data accompanying figure 4b can be found at (Friddle and Thürmer 2019b).

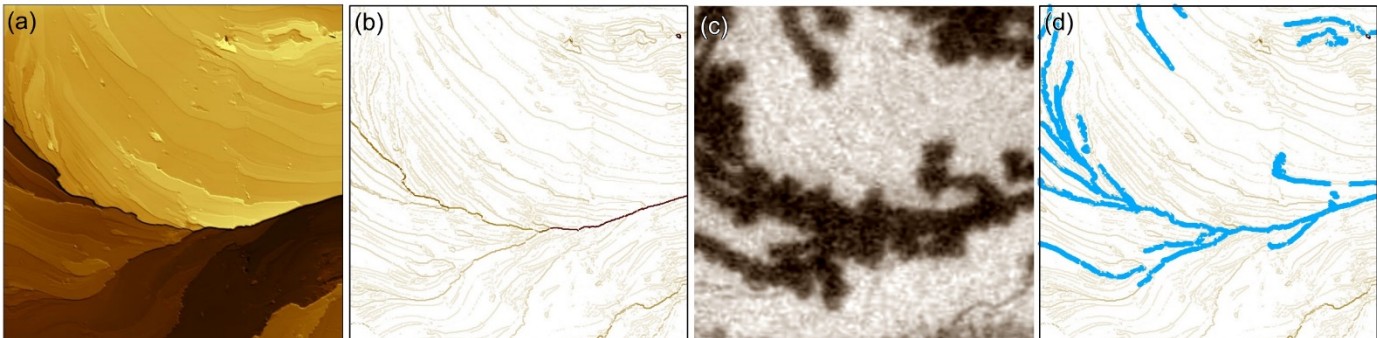

**Figure 5:** Processing video and AFM data into step-height statistics. An AFM image **(a)** is converted to an image of step-heights **(b)** and compared to the corresponding optical image of ice formation **(c)**. The step-height image in panel (b) is interactive in that the traces corresponding to ice-covered step segments can be selected by mouse click, shown as blue traces in (d).

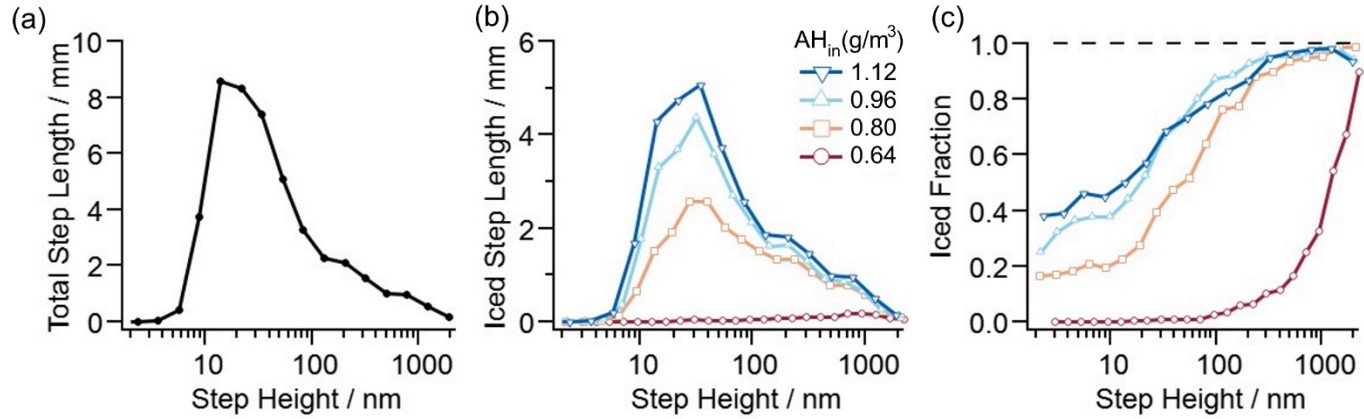

**Figure 6:** Step-height statistics. **(a)** Histogram of *all* step heights as measured by the total length of *all* steps within a step-height bin. **(b)** Histogram of step heights as measured by the total length of *iced* steps within a step-height bin. The histograms evaluate 15 images, each image covering $100 \times 100$ µm$^2$. The bins are logarithmically spaced, and binning for each step height is counted up as the sum of step-edge lengths. **(c)** Probability of finding ice on a step of a given height, computed as the total length of iced steps within a step-height bin divided by the total length of *all* steps (within the same step-height bin). The data show that increasing humidity shifts the distribution to the left, thus increasing the probability of finding ice on smaller step heights. Fluctuations in the ice fraction curves, particularly at high $AH_{in}$ and large steps, reflect the random dehydration of some larger step heights which are present in far fewer numbers than smaller step heights.