# Peer review of "Mapping ice formation to mineral-surface topography using a micro mixing chamber with video and atomic-force microscopy"

_Atmospheric Measurement Techniques, 2019_

## Editor Comment (EC1) · Mingjin Tang (Editor) · 24 Sep 2019

As the handling editor, I post this comment to explain what happened to this manuscript during the access review. The original manuscript was assessed by three referees during the access review stage. One referee stated that the work had one fundamental flaw, and therefore I initially decided to reject this manuscript. The authors contacted me via email, stating that lack of clarity in the original manuscript led to misunderstanding by the referee, and they also sent me the revised manuscript. What the authors stated appears to be sensible. At that point I could insist on my initial decision and asked the authors to resubmit their work as a "new" manuscript. Nevertheless, I de-

cided to accept the revised manuscript for open discussion due to reasons given below: 1) it may not be necessary for the authors and referees to start from the very beginning again; 2) the work itself seems to be novel, though major revision may be needed; 3) after the manuscript is online for open discussion, it will be rigorously reviewed. I would like to thank the three referees involved in the access review, and hope that they will provide full reviews after the manuscript is online for open discussion. In addition, colleagues interested in this work can also post their comments.

---

## Referee Comment (RC1) · Anonymous Referee #1 · 19 Oct 2019

Review for Mapping ice formation to mineral-surface topography using a micro mixing chamber with video and atomic-force microscopy by Raymond W. Friddle and Konrad Thürmer

I would like to congratulate the authors on the development of a novel and useful instrument for linking topographic features on bulk systems to ice growth. The results are clearly presented and the manuscript is well written. Nevertheless, I have a few comments listed below.

General comments:

The paper initially describes the technique as an instrument for elucidating atmospheric

ice formation. However, the primary focus of the results are about ice growth and propagation on the feldspar mineral. Although this is an interesting observation and result, it is not very atmospherically relevant. As in the atmosphere, the aerosols acting as INPs are between approximately 50 and 10000 nm. Therefore, it is likely that individual droplets would not exist on the surface of the aerosol particle. Rather, the entire aerosol would be immersed in a cloud droplet above water saturation and the ice nucleation event would cause the entire droplet to freeze. This renders the step height analysis unnecessary for atmospheric ice formation. I think this should be more clearly presented in the manuscript.

Nevertheless, the step height analysis is potentially an interesting and important result for the material science, biomedical and food preservation fields. Perhaps the authors should present the step height analysis in reference to those fields.

Although it is discussed that certain sites repeatedly nucleated ice while others lost that ability, it would be nice to show some examples of the types of sites that retained or lost their ice nucleating ability. For example, do they differ in geometry, location on the mineral surface etc.

Do the crystals that emerge from pits below water saturation or protrusions above water saturation have the same orientation as discussed in the Kiselev et al., (2017) study?

Minor Comments What is the resolution of the AFM? What is the tip width and how does this affect the mapping of the topographic features?

What is the temperature uncertainty of the thermistor? Is there an impact of the temperature measurement occurring below the standoff stage rather than below the sample itself (see Fig. 1)?

What are the uncertainties in the iced step height analysis? Please add error bars to the Fig. 6. Is there a reason that the largest step heights have a lower iced fraction above 0.75 AHin or is this due to the uncertainty of calculating the iced fraction of a

step. This result is in direct conflict with the statement that higher iced steps would retain ice longer than shorter steps (see discussion and outlook).

It is not stated how the humidity would be calibrated at other temperatures? Would the AHin be increased until water is observed and then this be used as 100 % RH in the cell?

I understand that once droplets are formed, the humidity would drop in the chamber, but at the highest AHin used in the study ∼200 % RH, do the droplets continue to grow/merge?

As mentioned in the general comments, the experiments conducted above water saturation are investigating ice growth. Please make this clearer on page 5 line 11.

Detailed comments Page 2 Line 5: Please add Pach and Verdaguer, (2019)

Page 2 line 16: Remove "however" as this confuses the sentence and move the citations to the end of the sentence.

Section 2.1.1 please reference Figure 1.

Section 2.1.1 on page 3 line 10 is not numbered correctly. Please change to 2.1.2

Page 3 line 16: Capitalize "figure"

Page 3 line 24-26: This sentence seems unnecessary here especially as it is not explored in this study. Ether reformulate to state that in theory this would be an additional advantage or remove.

Page 5 line 3: Please add appropriate references for ice formation from capillary condensation such as: Campbell et al., (2017); Campbell and Christenson, (2018); David et al., (2019); Marcolli, (2014); Pach and Verdaguer, (2019)

Page 6 line 5: numbering of section is off, change to 4.2.

Page 6 line 27: please change ice formation to ice growth

References:

Campbell, J. M. and Christenson, H. K.: Nucleation- and Emergence-Limited Growth of Ice from Pores, Phys. Rev. Lett., 120(16), 165701, doi:10.1103/PhysRevLett.120.165701, 2018.

Campbell, J. M., Meldrum, F. C. and Christenson, H. K.: Observing the formation of ice and organic crystals in active sites, Proc. Natl. Acad. Sci., 114(5), 810–815, 2017.

David, R. O., Marcolli, C., Fahrni, J., Qiu, Y., Sirkin, Y. A. P., Molinero, V., Mahrt, F., Brühwiler, D., Lohmann, U. and Kanji, Z. A.: Pore condensation and freezing is responsible for ice formation below water saturation for porous particles, Proc. Natl. Acad. Sci., 116(17), 8184–8189, doi:10.1073/pnas.1813647116, 2019.

Kiselev, A., Bachmann, F., Pedevilla, P., Cox, S. J., Michaelides, A., Gerthsen, D. and Leisner, T.: Active sites in heterogeneous ice nucleation—the example of K-rich feldspars, Science, 355(6323), 367–371, doi:10.1126/science.aai8034, 2017.

Marcolli, C.: Deposition nucleation viewed as homogeneous or immersion freezing in pores and cavities, Atmos Chem Phys, 14(4), 2071–2104, doi:10.5194/acp-14-2071-2014, 2014.

Pach, E. and Verdaguer, A.: Pores Dominate Ice Nucleation on Feldspars, J. Phys. Chem. C, 123(34), 20998–21004, doi:10.1021/acs.jpcc.9b05845, 2019.
* * *

---

## Referee Comment (RC2) · Anonymous Referee #2 · 22 Oct 2019

**Referee report on "Mapping ice formation to mineral-surface topography using a micro mixing chamber with video and atomic-force microscopy" by Raymond W. Friddle and Konrad Thürmer**

The authors have assembled an experimental setup to grow ice crystals on a sample surface and developed a method to locate where ice forms to investigate the topographical features on the underlying surface by atomic-force microscopy. While the setup could prove useful to study ice growth on surfaces, the intended use to study ice nucleation mechanisms requires a higher vertical resolution to detect small ice crystals and pinpoint the location of ice active sites. In addition, much better control of temperature and relative humidity in the mixing chamber is needed. I think such could be achieved and encourage the authors to improve the setup towards this direction.

Specific comments

Page 1 line 16 Heterogeneous ice nucleation is not limited to temperatures above -36°C. Deposition ice nucleation relevant for cirrus cloud formation occurs at lower temperatures and below water saturation. As the experimental setup described in this manuscript might become useful to investigate deposition ice nucleation, I recommend mentioning it here in the introduction.

Page 1 line 24 The parametrization by DeMott et al., 2010 is not based on size as an ice nucleation property, but simply relates the concentration of INP to the concentration of particles above a threshold size, not implying that only these particles act as INP. This is often misinterpreted, please revise.

Page 1 line 26 Useful parametrizations should capture various situations. Please elaborate and provide references supporting the claim that the mentioned parametrizations are not accurate outside the conditions for which they were developed. Also, surface site density of ice active sites derived from field measurements and laboratory studies have been used to parameterize ice formation in models eg., Vergara-Temprado et al., 2017. This could be mentioned.

Page 1 line 31 Please specify what kind of information microscopy can provide to distinguish mechanisms of ice nucleation.

Page 2 line 3f It is unclear how the 10nm size is derived. Given the resolution of light microscopy, pixel size etc., used in the current setup it seems unrealistic to detect such small objects, making the discussion of framerate and its dependence on temperature and humidity conditions irrelevant. What is the smallest detectable size in the current setup and what is the limiting component?

Page 2 line 15 Clarify how this estimate was made. The resolution is 1.6um? This seems not to be high enough to see growth of 1um crystals. In addition, I calculate at least 10-times longer growth needed to reach this size at this conditions. The mentioned growth rate indicates a RH>>100% and questions the control of relative humidity in the experiment. Ice growth can be used to infer humidity in the specimen chamber (see S3 in Kiselev et al., 2016). I highly recommend a comparison of relative humidity based on ice growth rates and the method used by the authors to determine humidity.

Page 2 line 16f Please elaborate how high-speed AFM can advance heterogeneous nucleation research.

Page 2 line 20 How accurate can the site of ice formation be located with this setup? It is mentioned on page 2 line 1 that the spatial resolution must be on the order of nanometers to locate the ice nucleation site. Please derive the minimum resolved distance for your camera system and verify with

a resolution target. A discussion of what accuracy would be desirable in contrast to what can be achieved would be helpful to clarify down to what scale the setup can be sensitive.

Page 2 line 22 Surface features on a feldspar specimen of the size used in this study might not be present on micrometre sized dust particles found at mixed-phase cloud level, and therefore be not relevant for ice nucleation on these particles. I recommend not to emphasise atmospheric relevance.

Page 3 line 1ff Provide a temperature calibration to demonstrate the stability (1°C/hr mentioned in Sec.2.2.), accuracy of temperature control and homogeneity in the mixing chamber. Temperature control is crucial to study ice nucleation and therefore the interpretation of observations made with the setup. Please clarify if temperature is actively controlled or only monitored with the TC-720. Active temperature control is desirable for this type of setup.

Page 3 line 11 Ice and mixed-phase clouds form at a variety of conditions. Ice clouds do not require water saturated conditions. Specify conditions that can be crated in the mixing chamber.

Page 3 line 17f How is frost formation in the mixing column prevented?

Page 3 line 20f Advantages compared to what other technique? What can be learned from using different flow rates?

Page 3 line 23f Please explain why thermal gradients are minimized by that.

Page 3 line 32 Please provide exemplary time series of temperature and relative humidity during an experiment. What is the purpose of switching the wet flow on and off? Could the humidity sensor be used to measure humidity in the outlet flow to verify the humidity in the chamber?

Page 4 line 1 How long does it take to reach steady-state humidity? To vary the humidity in the mixing chamber the flow through the bubbler is adjusted. Does this change steady state? Provide measured humidity after the bubbler as function of flow rate. Another strategy to adjust humidity in the wet flow might be to change the temperature of the bubbler.

Page 4 line 9ff Knowing and controlling the relative humidity (RH) in the experiment is essential for interpretation of results and to infer the ice nucleation mechanism. Calibration of relative humidity should be done much more carefully by eg., using a dew point mirror to measure humidity in the outflow of the chamber. While AH might be useful to determine flow rates of the wet flow, chamber conditions should be reported as relative humidity and temperature. Convert AH to RH throughout the manuscript.

Page 4 line 19 Converting the error in $AH_{in}$ of $0.08 g/m^3$ to RH gives +/- 18% which is a very high uncertainty for ice nucleation experiments.

Page 5 line 1 $AH_{in}$ reported here and considering the uncertainty given on the last page, relative humidity is equal to $RH_w$= 85% +/- 18%. Conditions above water saturation are within the experimental accuracy, making the interpretation of the data as purely deposition ice nucleation imprecise. This underlines the point made in the comment above, that control of the experimental conditions is insufficient for ice nucleation experiments. Compare estimated saturation conditions against calculation based on ice crystal growth rate or measure the humidity at the chamber outlet.

Page 5 line 3 Couldn't AFM detect pores on the substrate? What is the horizontal resolution of AFM used here?

Page 5 line 5 "Ice formation" instead of "ice nucleation" would be more accurate.

Page 5 line 12ff What is discussed here is ice growth and not ice nucleation. Inferring ice nucleation mode from this observation seems over-reaching. The two processes (ice growth and ice nucleation) should be separated more clearly throughout the manuscript.

Page 6 line 14 All four humidities applied are high above water saturation (RH=134%, 167%, 201%, 234%). It is surprising to see sensitivity of ice formation on the amount of supersaturation in this high humidity regime other than a change in growth rate. As pointed out in the discussion, different grow rates are a more plausible explanation for the observation than the probability of ice nucleation. The context in which the experimental results are interpreted should be clarified. Is it about ice growth or ice nucleation mechanisms?

Page 7 line 5 Please provide the resolution of the current setup. Is the CCD pixel size limiting the resolution?

Page 10 Fig.2 check if there is a mix-up between e), d). The description in the figure caption seems to be switched. Images show a scale bar of 5um and this seems to be a typical scale how accurate ice formation can be located. In the introduction it is correctly mentioned that ice nucleation occurs on structures with a scale of few nanometres. Features in eg. e) are on a 1000-times larger scale, questioning the interpretation as ice nucleating sites.

Page 10 Fig. 3 replace AH with RH (=167% +/- 18%).

Page 12 Fig. 6 b) replace AH with RH  (=134%, 167%, 201%, 234% +/-18%) .

References

DeMott, P. J., Prenni, A. J., Liu, X., Kreidenweis, S. M., Petters, M. D., Twohy, C. H., Richardson, M. S., Eidhammer, T., and Rogers, D. C.:Predicting global atmospheric ice nuclei distributions and their impacts on climate, Proceedings of the National Academy of Sciences, 107, 11 217–11 222, https://doi.org/10.1073/pnas.0910818107, 2010.

Kiselev, A., Bachmann, F., Pedevilla, P., Cox, S. J., Michaelides, A., Gerthsen, D., and Leisner, T.: Active sites in heterogeneous ice nucleation-the example of K-rich feldspars, Science, 355, 367–371, https://doi.org/10.1126/science.aai8034, 2016.

Vergara-Temprado, J., Murray, B. J., Wilson, T. W., O'Sullivan, D., Browse, J., Pringle, K. J., Ardon-Dryer, K., Bertram, A. K., Burrows, S. M., Ceburnis, D., DeMott, P. J., Mason, R. H., O'Dowd, C. D., Rinaldi, M., and Carslaw, K. S.: Contribution of feldspar and marine organic aerosols to global ice nucleating particle concentrations, Atmos. Chem. Phys., 17, 3637–3658, https://doi.org/10.5194/acp-17-3637-2017, 2017.

---

## Author Comment (AC1) · 18 Dec 2019

**Author's response to Anonymous Referee #1**
Below we provide in blue-colored font a point-by-point reply to each comment.

I would like to congratulate the authors on the development of a novel and useful instrument for linking topographic features on bulk systems to ice growth. The results are clearly presented and the manuscript is well written. Nevertheless, I have a few comments listed below.

We thank the referee for the kind words regarding our novel approach to studying ice growth, as well as the referee's insightful comments.

General comments:
The paper initially describes the technique as an instrument for elucidating atmospheric ice formation. However, the primary focus of the results are about ice growth and propagation on the feldspar mineral. Although this is an interesting observation and result, it is not very atmospherically relevant. As in the atmosphere, the aerosols acting as INPs are between approximately 50 and 10000 nm. Therefore, it is likely that individual droplets would not exist on the surface of the aerosol particle. Rather, the entire aerosol would be immersed in a cloud droplet above water saturation and the ice nucleation event would cause the entire droplet to freeze. This renders the step height analysis unnecessary for atmospheric ice formation. I think this should be more clearly presented in the manuscript.

We concur with the referee that our observations on extended feldspar surfaces cannot straightforwardly applied to atmospheric conditions.  We agree that most aerosol particles are typically completely immersed in a cloud droplet already at very modest supersaturations. We address this issue by adding text in the manuscript to: (1) Clarify at the outset that and why we are looking at extended substrates, thus not raising the not-to-be-fulfilled expectation that we examine realistic aerosol particles. (2)  Describe more precisely the atmospheric conditions for which we believe our results might be relevant. Specifically, we:
Changed the first sentence in the abstract to: "*We developed a method for examining ice formation on solid substrates exposed to cloud-like atmospheres.*"
Before the last sentence of the introduction we added: "*While studying aerosol particles collected from the atmosphere would provide a more direct connection to atmospheric conditions. The typically complex structure and chemistry of these particles often precludes identifying the individual nanoscale processes that are important.  For this study, instead, we choose extended flat substrates of known composition on which the role of individual topographic features can be examined.*"
In first sentence of paragraph 2.1.1. changed "*To create cloud-like conditions*"  to "*To create a cloud-like atmosphere*".
In the discussion and outlook section we added the following two statements:
"*This process can be viewed as an extension of the pore condensation and freezing mechanism (Christenson 2013; David et al. 2019; Fukuta 1966; Marcolli 2014; Pach and Verdaguer 2019) to higher humidity.*"
" Typically, most aerosol particles are completely immersed in a cloud droplet already at very modest supersaturations.  As discussed in detail in (Friddle and Thürmer 2019a), the step-facilitated mechanism described above is expected to be relevant when a cavity-free feldspar particle, initially devoid of ice, is suspended in air colder than -20$^o$C that becomes slowly saturated.  According to Fletcher's estimate (Fletcher 1962; Pruppacher and Klett 1997) that a humidity of RH$_w$> 130% is required for a measurable nucleation rate of water droplets with a contact angle of ≈ 45$^o$ on a planar insoluble substrate.  Hence

*condensation of supercooled water will be confined to step edges, where the water will freeze rapidly, thus initiating ice formation."*

Nevertheless, the step height analysis is potentially an interesting and important result for the material science, biomedical and food preservation fields. Perhaps the authors should present the step height analysis in reference to those fields.

Following the referee's suggestion we added to following statement to the discussion and outlook section: "*In the discussed example of ice formation, the step-height analysis is used to corroborate the involvement of the liquid phase of water during the observed rapid formation and propagation of ice on feldspar, while the link between surface-step height and the ability of an isolated aerosol particle to initiate ice nucleation is neither direct nor obvious. Nevertheless, such step-height analysis might benefit future studies in fields of material science, like corrosion and aircraft icing (Gent, Dart, and Cansdale 2000; Kreder et al. 2016), where the behavior of the examined materials is affected by the abundance of surface steps.*"

Although it is discussed that certain sites repeatedly nucleated ice while others lost that ability, it would be nice to show some examples of the types of sites that retained or lost their ice nucleating ability. For example, do they differ in geometry, location on the mineral surface etc.

We did not perform an exhaustive study on the sites that lose or gain the ability to nucleate ice. With our optically limited spatial resolution of the ice crystals we are unable to decisively make a statement on the local surface structure (sub 10 nm) where nucleation occurs.

Do the crystals that emerge from pits below water saturation or protrusions above water saturation have the same orientation as discussed in the Kiselev et al., (2017) study?

We did not observe the regularity in crystal orientation as reported in the Kiselev study. We added at the end of section 4: "Our data neither reveal nor rule out any preferred crystal orientation of the observed ice structures."

Minor Comments:
What is the resolution of the AFM? What is the tip width and how does this affect the mapping of the topographic features?

We did not characterize the radii of the AFM tips we used. Although the manufacturer specifies the tip radii is < 10 nm, typically over extended use situations the tip radius is approximately 20 nm, putting the theoretical lateral resolution at about 4nm, while the vertical resolution is atomic. That said, the data used in our mapping is constructed of individual scans of 512x512 pixels covering 100x100 $\mu m^2$. This limits the spatial resolution to the pixel size of 195 nm. We included the tip model and vendor in section 2.2.

What is the temperature uncertainty of the thermistor? Is there an impact of the temperature measurement occurring below the standoff stage rather than below the sample itself (see Fig. 1)?

The inherent error in the thermistor reading is negligible (less than 0.1 C) compared to the uncertainty introduced by the sensor placement below the sample. In the setup in for these experiments we do not

use a sensor within the sample volume, however the flux of cold gas impinging the base of the sample plate is 20 times greater than the flux flowing over the top of the sample. Thus, only a small differential is expected between the sample base and sample volume temperatures. We revisited our temperature measurements and found a slight error in our original reporting of the temperature of the sample plate. Our measurements of the sample plate temperature were on average -29.5 ± 0.2 °C, where the error is the standard deviation of the readings over 9 runs. We take a conservative estimate to place the error at ± 0.5 °C. We have included this revised temperature and error throughout the paper, as well as adjusted the estimated AHin values accordingly.

What are the uncertainties in the iced step height analysis? Please add error bars to the Fig. 6. Is there a reason that the largest step heights have a lower iced fraction above 0.75 AHin or is this due to the uncertainty of calculating the iced fraction of a step. This result is in direct conflict with the statement that higher iced steps would retain ice longer than shorter steps (see discussion and outlook).

Uncertainty in histogram data is difficult to define without a priori knowledge of the distribution, and therefore the variance of a given bin. The noise observed in the curves for large humidities at large step heights arises from the random nature of ice coverage and limited data. The order of step icing, and in turn local dehydration, is random in each experiment. Therefore, in some runs a section of steps will be dehydrated, while in others those same steps will become iced. Since tall step heights are present in fewer numbers this random dehydration path can remove a noticeable portion of those steps from the overall counts of iced steps. We included the following sentence in the caption to figure 6, "Fluctuations in the ice fraction curves, particularly at high AHin and large steps, reflect the random dehydration of some larger step heights which are present in far fewer numbers than smaller step heights."

It is not stated how the humidity would be calibrated at other temperatures? Would the AHin be increased until water is observed and then this be used as 100 % RH in the cell?

The reviewer is correct, the humidity would have to be re-calibrated for a different temperature.

I understand that once droplets are formed, the humidity would drop in the chamber, but at the highest AHin used in the study ~200 % RH, do the droplets continue to grow/merge? As mentioned in the general comments, the experiments conducted above water saturation are investigating ice growth. Please make this clearer on page 5 line 11.

The actual humidity within the cell volume local to the viewing area is unknown and must be well below 200 % RH. As seen in the accompanying videos, in most areas the ice forms before the droplets are able to grow/merge to an appreciable size. We have changed this sentence to read, "Above saturation, we observe a very different pathway to ice formation." Where we removed "mode of" to distinguish from direct observation of ice nucleation.

Detailed comments:
Page 2 Line 5: Please add Pach and Verdaguer, (2019)    We have now added this reference twice in the experimental results section 3.
Page 2 line 16: Remove "however" as this confuses the sentence and move the citations to the end of the sentence.  Thank you for the suggestion, we have made this change.
Section 2.1.1 please reference Figure 1.  Thank you, this is called out in the first sentence of section 2.1.1.

Section 2.1.1 on page 3 line 10 is not numbered correctly. Please change to 2.1.2  Thank you, we have corrected this.
Page 3 line 16: Capitalize "figure"   Thank you, we have corrected this.

Page 3 line 24-26: This sentence seems unnecessary here especially as it is not explored in this study. Ether reformulate to state that in theory this would be an additional advantage or remove.

We reformulated this sentence to: "The *last point has the potential benefit of keeping the AFM tip near the same temperature as the sample surface, allowing, in principle, to image ice or the sample surface without raising their temperature (not explored here).*"

Page 5 line 3: Please add appropriate references for ice formation from capillary condensation such as: Campbell et al., (2017); Campbell and Christenson, (2018); David et al., (2019); Marcolli, (2014); Pach and Verdaguer, (2019)

Thank you, we have added the following appropriate references here: (Christenson 2013; David et al. 2019; Fukuta 1966; Marcolli 2014; Pach and Verdaguer 2019)

Page 6 line 5: numbering of section is off, change to 4.2. Thank you, we have corrected this.

Page 6 line 27: please change ice formation to ice growth   We have made the change.

References:
Campbell, J. M. and Christenson, H. K.: Nucleation- and Emergence-Limited Growth of Ice from Pores, Phys. Rev. Lett., 120(16), 165701, doi:10.1103/PhysRevLett.120.165701, 2018.
Campbell, J. M., Meldrum, F. C. and Christenson, H. K.: Observing the formation of ice and organic crystals in active sites, Proc. Natl. Acad. Sci., 114(5), 810–815, 2017.
David, R. O., Marcolli, C., Fahrni, J., Qiu, Y., Sirkin, Y. A. P., Molinero, V., Mahrt, F., Brühwiler, D., Lohmann, U. and Kanji, Z. A.: Pore condensation and freezing is responsible for ice formation below water saturation for porous particles, Proc. Natl. Acad. Sci., 116(17), 8184–8189, doi:10.1073/pnas.1813647116, 2019.
Kiselev, A., Bachmann, F., Pedevilla, P., Cox, S. J., Michaelides, A., Gerthsen, D. and Leisner, T.: Active sites in heterogeneous ice nucleationăĂ Tthe example of K-rich feldspars, Science, 355(6323), 367–371, doi:10.1126/science.aai8034, 2017.
Marcolli, C.: Deposition nucleation viewed as homogeneous or immersion freezing in pores and cavities, Atmos Chem Phys, 14(4), 2071–2104, doi:10.5194/acp-14-2071-2014, 2014.
Pach, E. and Verdaguer, A.: Pores Dominate Ice Nucleation on Feldspars, J. Phys. Chem. C, 123(34), 20998–21004, doi:10.1021/acs.jpcc.9b05845, 2019.

---

## Author Comment (AC2) · 18 Dec 2019

**Author's response to Anonymous Referee #2**
Below we provide in blue-colored font a point-by-point reply to each comment.

The authors have assembled an experimental setup to grow ice crystals on a sample surface and developed a method to locate where ice forms to investigate the topographical features on the underlying surface by atomic-force microscopy. While the setup could prove useful to study ice growth on surfaces, the intended use to study ice nucleation mechanisms requires a higher vertical resolution to detect small ice crystals and pinpoint the location of ice active sites. In addition, much better control of temperature and relative humidity in the mixing chamber is needed. I think such could be achieved and encourage the authors to improve the setup towards this direction.

We thank the reviewer for their detailed review of our manuscript and helpful comments.

We would like to clarify that our manuscript is intended to detail our approach of collocating ice formation/growth locations observed optically to the sample surface structure observed by AFM. If the interpretation is that we are monitoring ice formation with AFM, we apologize for the lack of clarity and emphasize that we only observe ice optically. We agree that with optical observation, which is ultimately wavelength-limited to at least a micron, it is impossible to pinpoint ice nucleation sites with the necessary resolution of less than 10 nm. Other approaches in the literature that have ostensibly studied ice nucleation have a similar resolution limit due to the risk of beam damage. The ESEM approach also does not control the humidity but infers the local humidity around a growing ice crystal through relating the ice growth rate to the local supersaturation. This calculation can be carried out in our work as well, but due to the network of ice which rapidly forms on the surface, the humidity varies substantially across the surface and with time. Determining a representative humidity at these length-scales is not trivial and is obviously not a number but a landscape. We respectfully contend that the main message of the paper stands without investigating the humidity in great detail at this stage in development. Afterall, we are not studying kinetics of ice nucleation or crystal growth where the exact supersaturation is necessary for drawing conclusions, but instead we are relating sample surface structure to a pathway of water vapor condensation and ice growth.

Specific comments
Page 1 line 16 Heterogeneous ice nucleation is not limited to temperatures above -36°C. Deposition ice nucleation relevant for cirrus cloud formation occurs at lower temperatures and below water saturation. As the experimental setup described in this manuscript might become useful to investigate deposition ice nucleation, I recommend mentioning it here in the introduction.

We have included this point in the discussion and outlook, "The setup described here is also applicable to studying deposition mode nucleation at sub -36 °C and sub-saturation (relevant to cirrus cloud formation), and optical observations of immersion mode nucleation on substrates."

Page 1 line 24 The parametrization by DeMott et al., 2010 is not based on size as an ice nucleation property, but simply relates the concentration of INP to the concentration of particles above a threshold size, not implying that only these particles act as INP. This is often misinterpreted, please revise.

We changed "based on macroscopic properties like aerosol size" to "relating the concentration of INPs to the concentration of particles above a threshold size".

Page 1 line 26 Useful parametrizations should capture various situations. Please elaborate and provide references supporting the claim that the mentioned parametrizations are not accurate outside the conditions for which they were developed. Also, surface site density of ice active sites derived from field measurements and laboratory studies have been used to parameterize ice formation in models eg., Vergara-Temprado et al., 2017. This could be mentioned.

We changed this sentence to "However, as is true for extrapolations in general, such models are expected to be less accurate when applied to conditions outside the range of measured values used to fine tune these models,", and included the reference to Vergara-Temprado et al.

Page 1 line 31 Please specify what kind of information microscopy can provide to distinguish mechanisms of ice nucleation.

By imaging the surface structure at an ice nucleation site, microscopy can provide additional information regarding condensation versus direct vapor deposition mechanisms if pores, cracks, or stepped structures are resolved or not. To further clarify this point we have modified the end of this sentence to read, "innovative microscopy aimed at uncovering the local structural and chemical properties of ice nucleation sites is needed.".

Page 2 line 3f It is unclear how the 10nm size is derived. Given the resolution of light microscopy, pixel size etc., used in the current setup it seems unrealistic to detect such small objects, making the discussion of framerate and its dependence on temperature and humidity conditions irrelevant. What is the smallest detectable size in the current setup and what is the limiting component?

This part of the paper presents an introduction to the challenges faced by using high resolution microscopy to study ice nucleation events. It is not a discussion of our approach, or our results. The 10 nm size is an order-of-magnitude benchmark based on the discussion in the first paragraph relating to critical-nucleus size. The value is used to exemplify the scale which non-optical microscopy techniques must be able to resolve, and at high frame rates, to truly observe ice nucleation at a definitive site.

Page 2 line 15 Clarify how this estimate was made. The resolution is 1.6um? This seems not to be high enough to see growth of 1um crystals. In addition, I calculate at least 10-times longer growth needed to reach this size at this conditions. The mentioned growth rate indicates a RH>>100% and questions the control of relative humidity in the experiment. Ice growth can be used to infer humidity in the specimen chamber (see S3 in Kiselev et al., 2016). I highly recommend a comparison of relative humidity based on ice growth rates and the method used by the authors to determine humidity.

We give an estimate of growth velocity for these structures, treating the projected area as an effective circle, only to serve as an example of the speeds involved in our observations of ice growth. Our growth rate is based on measurements made when the crystal is resolved optically (greater than 1um) then extrapolated to represent small length-scales. The growth habits we observe are polycrystalline, highly branched crystal clusters, whereas the velocity equation used by Kiselev et al., 2016 assumes an atomically flat spherical surface.

Page 2 line 16f Please elaborate how high-speed AFM can advance heterogeneous nucleation research.

HS-AFM can advance heterogeneous nucleation research by directly observing nucleating crystals with 10 nm resolution, or less, by way of high frame-rate AFM. This has the potential to yield the best nucleation site location accuracy of the techniques currently available. We added to the manuscript: "… if emerging ice crystals could be observed directly with ~10 nm resolution, thus improving significantly the accuracy of locating nucleation sites."

Page 2 line 20 How accurate can the site of ice formation be located with this setup? It is mentioned on page 2 line 1 that the spatial resolution must be on the order of nanometers to locate the ice nucleation site. Please derive the minimum resolved distance for your camera system and verify with a resolution target. A discussion of what accuracy would be desirable in contrast to what can be achieved would be helpful to clarify down to what scale the setup can be sensitive.

In our setup, we are ultimately limited by the wavelength of visible light used for optical microscopy. No light-based microscopy – including our own – would be sufficient to specify a nucleation site with any accuracy.

Page 2 line 22 Surface features on a feldspar specimen of the size used in this study might not be present on micrometre sized dust particles found at mixed-phase cloud level, and therefore be not relevant for ice nucleation on these particles. I recommend not to emphasize atmospheric relevance.

The morphological features we analyze in this paper are simple surface steps, which are ubiquitous and expected to be abundant on most if not all feldspar dust particles. However, the small size and the complex morphology of these aerosol particles makes it impossible to acquire high-resolution AFM data and isolate the role of individual surface steps. Also in response to reviewer #1, we added text to the manuscript intended to clarify the atmospheric relevance:

Before the last sentence of the introduction we added: "*While studying aerosol particles collected from the atmosphere would provide a more direct connection to atmospheric conditions. The typically complex structure and chemistry of these particles often precludes identifying the individual nanoscale processes that are important. For this study, instead, we choose extended flat substrates of known composition on which the role of individual topographic features can be examined.*"

And to the discussion and outlook section we added: "*In the discussed example of ice formation, the step-height analysis is used to corroborate the involvement of the liquid phase of water during the observed rapid formation and propagation of ice on feldspar…*"

Page 3 line 1ff Provide a temperature calibration to demonstrate the stability (1°C/hr mentioned in Sec.2.2.), accuracy of temperature control and homogeneity in the mixing chamber. Temperature control is crucial to study ice nucleation and therefore the interpretation of observations made with the setup. Please clarify if temperature is actively controlled or only monitored with the TC-720. Active temperature control is desirable for this type of setup.

The temperature control in this work is passive and monitored by the TC-720 controller. Typical experiments last less than 60 seconds (vapor flow on-to-off time). While we agree that strict temperature control is important when quantifying nucleation kinetics, we are not engaging those experiments here. The 1°C/hr stability figure was regularly observed during setup of experiments,

however it was not recorded. In lieu of this data we show the stability of the stage temperature over the minutes timescale for three cases: a) without vapor flow, b) during a short vapor flow experiment, and c) a long vapor flow experiment. In section 2.2 we have revised the paper to remove the mention of 1°C/hr, and instead include a discussion regarding the temperature variation during an experiment, "During the course of an icing experiment the temperature measured at the sample stage decreases by approximately 0.5 °C due to the diverting of a portion of cold gas back to the stage when the vapor flow is turned on."

[Figure]

Page 3 line 11 Ice and mixed-phase clouds form at a variety of conditions. Ice clouds do not require water saturated conditions. Specify conditions that can be crated in the mixing chamber.

Because the final humidity in the chamber is a mixture of vapor and dry flows, in principle the full range of humidities below saturation can be explored by adjusting the relative flow rates.

Page 3 line 17f How is frost formation in the mixing column prevented?

Over repeated experiments we do eventually form ice in the small mixing column portion of the cell. The small size of the column leads it to become clogged quickly once ice forms. This then leads to no flow into the cell. We remedied this by clearing out the cell with warm dry nitrogen between experiments.

Page 3 line 20f Advantages compared to what other technique? What can be learned from using different flow rates?

Control over total flow rate is useful in practice for various reasons. Large flow velocities can disrupt growing ice crystals that may be weakly bound to a substrate. On the other hand, if one were limited to slow flow rates this would cause long delays when changing cell conditions from, for example, low humidity to a desired humidity. The flow rates one might require will be substrate-dependent and also will depend on the conditions one is attempting to create within the cell (e.g. turbulent, laminar, etc.).

Page 3 line 23f Please explain why thermal gradients are minimized by that.

In a cold-stage only setup the gas above the sample is cooled by the stage. This results in a gradient in temperature from the gas-sample interface upwards. Here we have minimized this gradient by also flowing cold gas into the chamber so the sample is cooled from above and below.  To make this clearer we modified this sentence to "*In our setup,  a cold atmosphere flows into the small ~ 30 μL volume cooling the sample from above and below,  thus minimizing temperature differences laterally and from the sample-gas interface upwards.*"

Page 3 line 32 Please provide exemplary time series of temperature and relative humidity during an experiment. What is the purpose of switching the wet flow on and off? Could the humidity sensor be used to measure humidity in the outlet flow to verify the humidity in the chamber?

Page 4 line 1 How long does it take to reach steady-state humidity? To vary the humidity in the mixing chamber the flow through the bubbler is adjusted. Does this change steady state? Provide measured humidity after the bubbler as function of flow rate. Another strategy to adjust humidity in the wet flow might be to change the temperature of the bubbler.

The vapor flow is turned on and off to control the generation of ice during an experiment. We did not use the outlet to measure humidity because the humidity sensor time to steady state is long compared to the experiment. This is partly due to the capacitive sensor used for RH measurement as well as the volume around the sensor that must be exchanged. The humidity exiting the chamber over time is also convoluted by the ice growth within the small chamber volume, further compounding the inaccuracy of the final RH measurement. As an illustration, we include here the measured humidity at the outlet of the sample chamber at room temperature under similar conditions to typical experiments. Varying the flow rate of the bubbler changes the proportion of vapor mixed into the constant dry nitrogen flow. Changing the flow rate also changes the time with which steady state is reached. For example, in the data shown below, at the lowest bubbler flow rate the effective sensor time constant is approximately 45 seconds, while at the highest bubbler flow the time constant is about 20 seconds. These times are far too long for the typical duration of an icing experiment which is less than 1 minute.

[Figure]

Page 4 line 9ff Knowing and controlling the relative humidity (RH) in the experiment is essential for interpretation of results and to infer the ice nucleation mechanism. Calibration of relative humidity should be done much more carefully by eg., using a dew point mirror to measure humidity in the outflow of the chamber. While AH might be useful to determine flow rates of the wet flow, chamber conditions should be reported as relative humidity and temperature. Convert AH to RH throughout the manuscript.

We opted to provide the estimated AH input to the cell rather than a RH because the RH is unknown at the surface location where we observe the ice formation/growth. Since we are not calculating quantitative kinetic parameters related to ice nucleation we show here the trend for increasing humidity without declaring that we have accurate knowledge of the actual humidity where the events take place.

Page 4 line 19 Converting the error in AHin of 0.08g/m3 to RH gives +/- 18% which is a very high uncertainty for ice nucleation experiments.

Please see above.

Page 5 line 1 AHin reported here and considering the uncertainty given on the last page, relative humidity is equal to RHw= 85% +/- 18%. Conditions above water saturation are within the experimental accuracy, making the interpretation of the data as purely deposition ice nucleation imprecise. This underlines the point made in the comment above, that control of the experimental conditions is insufficient for ice nucleation experiments. Compare estimated saturation conditions against calculation based on ice crystal growth rate or measure the humidity at the chamber outlet.

Please see above.

Page 5 line 3 Couldn't AFM detect pores on the substrate? What is the horizontal resolution of AFM used here?

The AFM can routinely detect features down to sub-10 nm in size. However, the optical and temporal resolution of our camera is not adequate to observe an ice nucleation event – it can only record the subsequent growth of the ice crystals. Therefore, any attempt to associate a point on an AFM image with an ice crystal's nucleation site will have an error of several microns.

Page 5 line 5 "Ice formation" instead of "ice nucleation" would be more accurate.

We agree with the referee and we have changed this in the text.

Page 5 line 12ff What is discussed here is ice growth and not ice nucleation. Inferring ice nucleation mode from this observation seems over-reaching. The two processes (ice growth and ice nucleation) should be separated more clearly throughout the manuscript.

We agree and have changed this from "mode" to "pathway".

Page 6 line 14 All four humidities applied are high above water saturation (RH=134%, 167%, 201%, 234%). It is surprising to see sensitivity of ice formation on the amount of supersaturation in this high humidity regime other than a change in growth rate. As pointed out in the discussion, different grow rates are a more plausible explanation for the observation than the probability of ice nucleation. The context in which the experimental results are interpreted should be clarified. Is it about ice growth or ice nucleation mechanisms?

The RH at the surface is clearly not given by the numbers mentioned above, otherwise the cell would be full of water. This is why we provide the input AH to convey the trend in humidity without claiming precise knowledge of the actual humidity at the surface. The benefit of using humidity greater than

saturation is that it reveals very clearly the ability of surface steps to provide pore-like condensation channels. If we limited our experiments to RH at saturation, the limited water content in the channels would be quickly dehydrated once ice is formed nearby, making it difficult to observe the ubiquitous nature of the water channels on the surface. The larger context of our results is that by combining surface topography with optical microscopy we can better define the pathway to ice formation on surfaces.

Page 7 line 5 Please provide the resolution of the current setup. Is the CCD pixel size limiting the resolution?

We are limited to about 1.6 um resolution, which is partly a function of the NA of our long-working distance objective. We are not limited by the CCD pixel density.

Page 10 Fig.2 check if there is a mix-up between e), d). The description in the figure caption seems to be switched. Images show a scale bar of 5um and this seems to be a typical scale how accurate ice formation can be located. In the introduction it is correctly mentioned that ice nucleation occurs on structures with a scale of few nanometres. Features in eg. e) are on a 1000-times larger scale, questioning the interpretation as ice nucleating sites.

The scale bars here are correct. We modified the caption to "… *(e) Expanded view of a portion of panel (d)…"*. We did not intend to claim that we are able to pinpoint ice nucleation sites. We circle the area within which nucleation occurs and subsequently and ice crystal emerges. We do not claim to have isolated the precise site.

Page 10 Fig. 3 replace AH with RH (=167% +/- 18%).
Page 12 Fig. 6 b) replace AH with RH (=134%, 167%, 201%, 234% +/-18%) .

Please note the discussion above. We are currently unable to precisely define the RH at the surface and throughout the experiment, and therefore have opted to provide the approximate AHin for each run.

References
DeMott, P. J., Prenni, A. J., Liu, X., Kreidenweis, S. M., Petters, M. D., Twohy, C. H., Richardson, M. S., Eidhammer, T., and Rogers, D. C.:Predicting global atmospheric ice nuclei distributions and their impacts on climate, Proceedings of the National Academy of Sciences, 107, 11 217–11 222, https://doi.org/10.1073/pnas.0910818107, 2010.
Kiselev, A., Bachmann, F., Pedevilla, P., Cox, S. J., Michaelides, A., Gerthsen, D., and Leisner, T.: Active sites in heterogeneous ice nucleation-the example of K-rich feldspars, Science, 355, 367–371, https://doi.org/10.1126/science.aai8034, 2016.
Vergara-Temprado, J., Murray, B. J., Wilson, T. W., O'Sullivan, D., Browse, J., Pringle, K. J., Ardon-Dryer, K., Bertram, A. K., Burrows, S. M., Ceburnis, D., DeMott, P. J., Mason, R. H., O'Dowd, C. D., Rinaldi, M., and Carslaw, K. S.: Contribution of feldspar and marine organic aerosols to global ice nucleating particle concentrations, Atmos. Chem. Phys., 17, 3637–3658, https://doi.org/10.5194/acp-17-3637-2017, 2017.

---

## Author Response (AR2)

We thank the referee for his careful critique of our manuscript. We were pleased to read that the referee found, "The manuscript is well written and easy to follow." and also that, "The topic fits perfectly to the scope of the journal. The described approach further tackles a key question in atmospheric science presenting a possibility to relate surface structure to ice forming ability." We address the referee's broader and detailed comments below in blue Cambria font.

Referee #4: Thorsten Bartels-Rausch, thorsten.bartels-rausch@psi.ch

Suggestions for revision or reasons for rejection (will be published if the paper is accepted for final publication):

The manuscript by Friddle and Thürmer presents the development of a method to observe ice formation on well characterized surfaces under controlled temperature and humidity settings. The ice formation is then related to the observed surface features such as steps and defects.

The manuscript has two foci, it presents and characterized the novel set-up and it discusses whether or not ice formation is triggered on specific surface sites or not. I understand that this is an instrumental paper presenting a novel measurement approach. The discussion on ice formation serves as proof of concept. I therefore judge the manuscript mainly on how rigorous the set-up and procedure is described and tested. Also because, the novelty and uniqueness of the scientific discussion compared to the submitted manuscript Friddle 2019a is not possible to assess. That said, I find that the presentation of the set-up and approach lacks crucial information and supporting data.

The manuscript is well written and easy to follow. The length is appropriate while the level of details might be increased. The topic fits perfectly to the scope of the journal. The described approach further tackles a key question in atmospheric science presenting a possibility to relate surface structure to ice forming ability. What keeps me from accepting the manuscript in its current form is

• the missing detailed characterization of temperature and relative humidity in the cell. Temperature and water vapor pressure are crucial for ice nucleation and growth behavior and I fell that both issues require additional discussion and/or reference measurements.

Response #1
We agree that the paper will benefit from additional details regarding the temperature and humidity in the cell. We have addressed the referee's detailed comments in this regard below. Specifically, we've addressed the error in our measurements, variations in temperature during an experiment, and the range of temperature and humidity we can explore. We've also expanded the explanation of our sample cooling approach.
We also agree that variations in temperature and humidity in our setup are likely to affect rates of ice nucleation and growth. However, as we also address below in our responses to the referee's detailed comments, our manuscript does not attempt to quantify nucleation events or growth rates, but rather reports a pathway to condensation and freezing which is supported by our structural AFM data and the corresponding video microscopy. Variations in temperature and humidity that may exist in our case, clearly do not invalidate those experimental findings. In particular, the relationship between surface-step patterns and measured step heights on the one hand, and ice-formation patterns on the other hand, is robust enough not to be obscured by possible small variations in temperature and humidity.

• That I don't understand how observation of ice larger in size than the defects/surface features says anything about the ice nucleating or forming at the defect. It could just have nucleated next to the defect and grow over it. I think there might be an issue with resolution to draw sound conclusions. In this respect, I suggest to at least mention Knopf, npj Climate and Atmospheric Science, 2020 in the manuscript.

Response #2: The referee is right that, due to the limited resolution of optical microscopy, we cannot directly determine the nucleation site with nanometer precision, and we discuss these limitations in section 5 in the paragraph beginning "*In the current implementation of the AFM/optical technique presented in this paper….*", where we state, for example, that *"Ultimately, advanced high-speed AFM may be the key to direct observations of ice nucleation events,…"* Although we are not there yet, our approach does represent progress in that direction. By preparing cleavage surfaces that contain rather flat surface regions, we create configurations in which most steps are separated far enough to be optically resolved. This allows us to obtain unambiguous evidence relating ice-propagation patterns to surface steps, and to employ AFM to quantify the role of the steps' height. It is true that we do not have direct evidence that ice could not just have nucleated next to a step and have grown over it. However, the strong correlation of ice-formation patterns with surface-step patterns, in particular the persistence of the flattest regions (with fewer steps) to remain ice free until the ice spreads from the stepped regions, makes it implausible that steps do not facilitate ice formation. The mechanism for this facilitation we propose, and explain in more detail in (Friddle and Thürmer 2019a), is that at saturation (with respect to liquid water) the lower side of the steps will be filled with wedges of supercooled water, creating an interface between feldspar and supercooled water, where ice nucleation is more likely than at the feldspar-vapor interface. We thank the referee for pointing out the relevance of *(Knopf, npj Climate and Atmospheric Science, 2020)* to the discussion in our paper. We revised our manuscript to include part of the discussion above. We also now point out the connection to the reference by adding the following text to the discussion section: "*A step height-dependence of ice growth can be attributed to two causes. First, if the probability of a given body of supercooled water to freeze at any given moment is roughly proportional to the surface area of the feldspar substrate immersed in the supercooled water, consistent with the models for immersion freezing considered in (Knopf, npj Climate and Atmospheric Science, 2020 ), then the probability of a given step edge segment to initiate ice nucleation is roughly proportional to its height*".

Detailed comments:

Page 1, Line 6: "cloud like atmosphere": What is the operational range of your set-up both for temperature and for water vapour pressure that you have tested. Please specify the conditions.

The system is operated at ambient atmospheric pressure. By adjusting the relative flow rates we were able to achieve humidities between 0 and 90% RH at room temperature. The temperature can be controlled to as low -70C. We have now specified this in the last sentence of section 2.1.1, *"With this setup we are able to adjust the room-temperature $RH_w$ between 0 – 90 %, and the temperature of the sample stage to as low as -70 °C."*

Page 1, Line 9: "relate the likelihood of ice formation to nanoscale properties of a mineral substrate". This is true but misleading. Surface features can be resolved down to a nanometer scale, but the

smallest ice patch is a few um in diameter (Figure 6). I would strongly argue that the lower resolution determines the overall performance of the approach which would be um and not nanometer in this case. However, please specify the resolution of both methods in the abstract.

We completely agree with the fact that lateral resolution of ice nucleation is limited by our optical resolution. In this paper we do not claim to resolve actual nucleation sites. Instead, in the case of ice formation that is clearly templated by surface steps, we can quantify useful details about those steps at the nanometer length scale (specifically their step-height in this case). (See also our response #2 above).

Page 2, line 4 (and throughout the manuscript): relative humidity to what? Water or ice? Please specify.

RH with respect to water. Thank you for pointing this out, we have clarified this to read "$RH_w$" throughout the paper.

Page 2, line 16: "Meanwhile at RH …. Just 6 ms." Interesting fact about the set-up. Please move to the discussion and expand.

We agree with the referee's suggestions and we moved this sentence from the introduction, "A typical AFM scan of sub-micron size can take between 10 seconds to 10 minutes depending on flatness of the substrate, the field of view (FOV), and the desired resolution. Meanwhile at RHw ≈ 100 % and ≈ –30 °C we estimate that a new ice crystal reaches an effective diameter of 1 μm in just 6 ms ." and worked it into the second to last paragraph of the discussion/outlook as, *"Ultimately, advances in high-speed AFM (HS-AFM) may be the key to direct observations of ice nucleation events. A typical commercial AFM scan takes 10 seconds to 10 minutes depending on flatness of the substrate, the field of view (FOV), and the desired resolution. Meanwhile at RHw ≈ 100 % and ≈ –30 °C we estimate that a new ice crystal reaches an effective diameter of 1 μm in just 6 ms . Progress in HS-AFM development has proven the tool to be capable of performing fasthigh-speed imaging of dynamic processes at nanometer resolution under various environments (Yamashita et al. 2009; Payton, Picco, and Scott 2016; Pyne et al. 2009; Picco et al. 2008; Kodera et al. 2010; Uchihashi et al. 2011; Casuso et al. 2010)."*

Page 2 line 30: I miss a introduction to surface features/defects with focus on typical sizes and a discussion on which of these have been found to nucleate ice or foster the formation of ice.

We believe the reviewer is referring to Page 5 here. This portion of the paper briefly discusses our observations of ice which formed as isolated crystal clusters as opposed to the majority of ice which formed as extended filaments. The point here is that, in the few cases we observed, the surface under isolated ice clusters have short step edges on island-like terraces, whereas the surface under extended ice filaments have long running step edges. We did not do a complete study on surface features that lead to discrete ice crystal formation; however, these observations support the concept that ice is forming by freezing the pre-existing water condensed in step edges. That is, short step edges (like those found on small islands) will result in isolated clusters of ice because the source of water is limited to the short step edge length. This is discussed at the end of the first paragraph in section 5. To further clarify this discussion we have added *"step edges having short lengths, such as"* to the last sentence to read, *"Here, ice filaments grow when step edges maintained tall heights for extended distances, whereas isolated ice crystals were observed at step edges having short lengths, such as protrusions or depressions surrounded by flat areas."*

Page 3, line 10 "The humidity of the … ThermoWorks). Taken that relative humidity has such a profound impact on ice nucleation, growth and stability; I think that this short statement is not enough.

• What is the precision of the sensor?
• Do you achieve equilibrium vapor pressure in the bubbler filled with water? This could be achieved by dispersing smaller droplets; then you could calculate the RH based on the water temperature. From my experience this is much more reliable than sensors.
• Have you cross-checked the sensor with a dew-point sensor? That would be another option to verify.

The humidity sensor has an accuracy of ±3% $RH_w$ at 25 C, which is approximately where we operate the sensor (room temperature) at the outlet of the bubbler. We flow a stream of humid gas from the bubbler over the sensor for an extended period of time to reach a steady state humidity. It is this steady state humidity that flows into the chamber and mixes with cold nitrogen gas during an experiment. We emphasize that we provide the estimated absolute humidity (AH) input to the cell because the RH is unknown at the surface location where we observe the ice formation/growth. Since we are not calculating quantitative kinetic parameters related to ice nucleation, we show AH values for conveying the trend in increasing humidity without declaring that we have accurate knowledge of the actual humidity where the events take place. We have included the error of the sensor in section 2.1.1 as, *"The humidity of the resulting room-temperature vapor is measured using a humidity sensor (ThermaData Series II – HTF, ThermoWorks) with an accuracy of ±3% $RH_w$ at 25 °C."*

Page 3, line 13: What is real time? Please specify sample rate.
We have changed this sentence to explicitly state the sampling rate, *"… readings are sampled at 1 Hz using a TC-720 thermoelectric temperature controller…"*

Page 3, line 26: Which range of gas flows can be explored. Please specify.
We now specify the range of flow rates we explore by including the range in this sentence as, *"a range of flow rates can be explored (here 0.9 – 1.32 L/min),"*

Section 2.1.2
May I ask you to specify the temperature distribution and trend with time more detailed. The temperature at any location in the chamber is basically given by the cooling and heat flux. Isolating materials add complexity to this as their cooling rate is slow. Nevertheless, if you constantly cool your chamber with cold N2 gas, at some point the whole chamber will reach the temperature of the gas unless you actively heat from the outside. In your case, heating comes from the outside air. Given this complexity, I can't follow your arguments on why your set-up guarantees that the coldest spot is the sample. Can you specify and give some experimental proof. One possibility might be to set the partial pressure of water to lets' say 1 mbar which corresponds to the the vapour pressure of ice at -20°C, set T(sample) to -30°C to trigger ice growth, then to -20°C (or close to). Now, if you keep this running for hours, the ice will only remain or cover all parts that are at -20°C. Warmer parts will not be ice covered. Thus the spread of ice will show you which parts of the set-up are at -20°C and wich are warmer.

We do not claim that our setup guarantees the coldest spot is the sample. We only mention that in a cold stage approach, measures must be taken to ensure this. In our setup the sample is cooled by the cold N2 gas flowing into the chamber and also by a large flux of cold N2 gas impinging on the underside of the sample. In Figure 1b we show that the total flow rate of the cold N2 gas is 14.1 L/min, with 0.66 L/min of this stream used to flow into the chamber, and the remaining 13.44 L/min is directed at the underside of the sample puck. The cold gas flowing into the chamber is only 1/20 in flux than the gas cooling the underside of the sample, and the chamber gas mixes with a warmer humid gas upon starting the experiment. It is therefore reasonable to assume the sample puck and sample are at colder temperature than the rest of the cell. We measured the temperature under the sample puck and the temperature of the exiting chamber gas and find the chamber exit

gas reading to be about 5 to 6 C warmer than the puck, although some of this warming arises from the small exit tube that the gas passes through which exchanges any outside heat efficiently (Figure 1a). The objective in our setup is not to create a location that is the coldest, but to minimize temperature differences within the chamber in order to closer mimic realistic conditions.

We have included a new second paragraph in section 2.1.2 to provide these details,
"As mentioned in the previous section, the sample is cooled by the cold N2 gas entering the chamber and by cold N2 gas impinging on the underside of the sample. In our experiments, 0.66 L/min of the cold N2 flows into the chamber, while 13.44 L/min of cold N2 is directed at the underside of the sample puck.  Measurements of the temperature of the gas exiting the chamber find it to be about 5 to 6 C warmer than the reading directly under the sample puck. Some of the exit gas warming arises from exchanging heat with the small diameter outlet tube that the gas passes through before contacting the sensor (Figure 1a)."

Page 3, Line 28-31. All this is a result and should be moved to the results/discussion section.
We feel that this brief justification of our approach is more easily understood when following the explanation of the method and its comparison to other approaches.

Page 3 line 37. If you switch off the flow of water vapour, do you then not change the mixing ratio of cold to warm gas (as the water vapour flow is part of the warm gas) and thus change the temperature? Please specify the "consant temperature" in line 37.

By constant temperature we simply mean the temperature is not varied during the course of the icing experiment. Some small change in the temperature of the gas does occur due to mixing with the vapor. However we also have the strong flux of cold gas impinging under the sample which is not mixed with vapor. We have changed "constant" to "set" and a include a clearer explanation of the setting a few sentences later in the paragraph as,
*The temperature is not actively controlled, therefore a small change in the inlet gas temperature will occur when the vapor is mixed into the stream. During the course of an icing experiment the temperature measured at the chamber outlet warms by about 1 °C , while the sample stage cools by approximately 0.5 °C due to the diverting of a portion of cold gas back to the stage when the vapor flow is turned on."*

Page 4, line 14-16. "note that … values". This is correct and does not need the "can be" before the "much lower". Once you have ice in the chamber, its temperature determines the partial pressure of water in the cell which is always 100% relative to ice near the ice. If AHin is larger ,the ice growths.
We thank the reviewer for this helpful insight. We have changed "*can be*" to "*is*" in this sentence.

Page 4 line 17 – 20. I'm sorry, but I can't follow you here. Any onset of condensation might be kinetically hindered. Wouldn't it be better to perform this calibration at equilibrium conditions.

"Onset of condensation" is indeed inappropriate terminology.  Due to its kinetic limitations and its dependence on the strongly varying local surface morphology this onset cannot straightforwardly be associated with a fixed humidity value.  Also, due to the limited resolution of the optical microscope we cannot see the initial condensation occurring at step edges.  What we do instead is adjusting the humidity inlet flow until we see droplets of ~1-5 $\mu$m diameter neither grow nor shrink, until after a few seconds the first ice crystals appear, causing nearby droplet to disappear quickly.
To correct this misleading terminology, we changed the text (in the last paragraph of section 2.2) to
*"We estimate AH$_{in}$ by calibrating against the saturation (RH$_w$ ≈ 100 %) condition.  To establish the RH$_w$ ≈ 100 % condition, we adjust the humidity inlet flow until we see droplets of ~1-5 $\mu$m diameter*

*neither grow nor shrink, until after a few seconds the first ice crystals appear, causing nearby droplet to shrink and disappear. We then calculate the absolute humidity... "*

Page 5 line 5: Please detail why you expect liquid below 0°C. This comes a little abrupt and might sound odd to those not familiar with freezing point depression by inverse Kelvin effect which I assume you refer to here.

We address this by introducing the pore condensation and freezing properly. For this we rewrote the appropriate part of the first paragraph of section 3.

Page 6 line 25 "ice is more likely to form" should that not be ice is more likely to grow.
Discussion. Please discuss and mention temperature gradients along the sample. Can you rule out that those set the direction into which the ice grows or the locations where the ice forms. I'm convinced you can, but please specify.

To clarify our use of *"ice formation"* we inserted this footnote at the very beginning of the introduction: "*Throughout this manuscript we use the term "ice formation" for the entire process that produces ice, i.e., ice nucleation and growth and/or freezing."* At page 6 line 25 "ice...to form" refers specifically to the process shown in Fig. 4 and quantified in Figure 6c: capillary water condensation and freezing at the substrate steps. We describe this process briefly in this paragraph, which now also includes the connection to Ref. Knopf et al 2020. We kept this brief to avoid redundancy with Friddle and Thürmer 2019a, which is now available to the public.

Regarding temperature gradients, we added the following paragraph after the first paragraph of the discussion:

[revised manuscript text omitted]